# Biomechanical stress regulates mammalian tooth replacement via the integrin β1-RUNX2-Wnt pathway

Xiaoshan Wu[1,2], Jinrong Hu[3,4], Guoqing Li[1], Yan Li[1,5], Yang Li[1], Jing Zhang[1], Fu Wang[1,6], Ang Li[1,7], Lei Hu[1], Zhipeng Fan[1], Shouqin Lü[3,4], Gang Ding[1,8], Chunmei Zhang[1], Jinsong Wang[9], Mian Long[3,4] & Songlin Wang[1,9,*] iD

## Abstract

Renewal of integumentary organs occurs cyclically throughout an organism's lifetime, but the mechanism that initiates each cycle remains largely unknown. In a miniature pig model of tooth development that resembles tooth development in humans, the permanent tooth did not begin transitioning from the resting to the initiation stage until the deciduous tooth began to erupt. This eruption released the accumulated mechanical stress inside the mandible. Mechanical stress prevented permanent tooth development by regulating expression and activity of the integrin β1-ERK1-RUNX2 axis in the surrounding mesenchyme. We observed similar molecular expression patterns in human tooth germs. Importantly, the release of biomechanical stress induced downregulation of RUNX2-wingless/integrated (Wnt) signaling in the mesenchyme between the deciduous and permanent tooth and upregulation of Wnt signaling in the epithelium of the permanent tooth, triggering initiation of its development. Consequently, our findings identified biomechanical stress-associated Wnt modulation as a critical initiator of organ renewal, possibly shedding light on the mechanisms of integumentary organ regeneration.

**Keywords** biomechanics; organ replacement; stress; Wnt signaling
**Subject Categories** Development; Signal Transduction
**The EMBO Journal (2020) 39: e102374**

## Introduction

Cyclical renewal of various integumentary organs, including hair, feathers, and teeth, is necessary for maintaining their function after wear or injury (Blanpain & Fuchs, 2014; Lai & Chuong, 2016; Lu *et al*, 2016). Transition from the resting to the initiation stage is critical for organ replacement and regeneration. For example, determining how to induce hair follicles to make the transition from the telogen (resting stage) to the anagen phase (initiation stage) without falling into the exogen phase (shedding phase) may enable the development of alopecia treatments. Although it is known that circulating hormones and molecules play important roles in the transition from one phase to another (Fuchs, 2015; Widelitz & Chuong, 2016), the mechanism of organ renewal initiation remains to be identified.

The initiation of tooth development involves a series of interactions between the epithelium and the mesenchyme. It begins as the dental lamina (DL) invaginates into the mesenchyme and forms a tooth bud. After the epithelium folds into the mesenchyme, it grows into a cap and then enters the bell stage (Koussoulakou *et al*, 2009). Continuous tooth renewal occurs throughout the life span of many animals such as fish and reptiles (Huysseune, 2006; Handrigan & Richman, 2010; Handrigan *et al*, 2010; Wu *et al*, 2013). However, the teeth of large mammals—including humans—are replaced only once (Jarvinen *et al*, 2009; Jernvall & Thesleff, 2012). The dental lamina of permanent teeth (PT) can be detected at a very early embryonic stage, but the development of PT germs continues for about 6–12 years before eruption (Pansky, 1982). It is known that the dental lamina remains in the resting phase for a while before developing into a tooth bud (Fraser *et al*, 2006). However, little is known about how the resting dental lamina of the PT is initiated

---

1 Beijing Key Laboratory of Tooth Regeneration and Function Reconstruction, Capital Medical University School of Stomatology, Beijing, China
2 Department of Oral and Maxillofacial Surgery, Xiangya Hospital, Central South University, Changsha, China
3 Center of Biomechanics and Bioengineering, Key Laboratory of Microgravity (National Microgravity Laboratory) and Beijing Key Laboratory of Engineered Construction and Mechanobiology, Institute of Mechanics, Chinese Academy of Sciences, Beijing, China
4 School of Engineering Science, University of Chinese Academy of Sciences, Beijing, China
5 Fortune Link Triones Scitech Co., Ltd., Beijing, China
6 Department of Oral Basic Science, School of Stomatology, Dalian Medical University, Dalian, China
7 Key Laboratory of Shaanxi Province for Craniofacial Precision Medicine Research, College of Stomatology, Xi'an Jiaotong University, Xi'an, China
8 Department of Stomatology, Yidu Central Hospital, Weifang Medical University, Weifang, China
9 Department of Biochemistry and Molecular Biology, Capital Medical University School of Basic Medical Sciences, Beijing, China
*Corresponding author. Tel:+86 010 57099478, E-mail: slwang@ccmu.edu.cn

and regulated inside the mandible, largely because of the lack of suitable animal models.

The miniature pig has a diphyodont dentition similar to that of humans. Thus, our group established a miniature pig model as a tooth development research platform (Wang *et al*, 2014, 2017; Li *et al*, 2015, 2018). Studies using large animal models have defined the developmental stages and established gene expression profiles of diphyodont dentition (Weaver *et al*, 1966; Tucker & Widowski, 2009; Buchtova *et al*, 2012; Sova *et al*, 2018). In this study, we investigated the mechanism by which tooth replacement is initiated using both the miniature pig model and human canine tooth germ samples.

# Results

## Morphology of the permanent pig canine tooth during the initiation stage

We selected deciduous and permanent canine teeth as a model for observing the early stages of PT morphogenesis (Fig 1A–G). The successional dental lamina (SDL) of the permanent canine (PC) bud was connected to the outer enamel epithelium of the enamel organ of the deciduous canine (DC) on the lingual side at embryonic day 50 (E50) (Fig 1A). At E55, the SDL began to separate from the DC (Fig 1B). Hematoxylin and eosin (H&E) staining and immunostaining showed that the SDL contained medial and lateral layers (Fig EV1A and B). At E60, the SDL of the PC had separated from the DC. A space could be identified between the DC and PC (Fig 1C). The SDL remained resting until the start of the bud stage, when the DC started to erupt (Fig 1D and E). At E90, the DC had erupted and the enamel organ of the PC at the bud stage was localized beside the gingival sulcus near the top (Fig 1E). At postnatal day 0 (PN0), the PC had entered the cap stage (Fig 1F); at PN10, it had entered the bell stage (Fig 1G). To observe the morphological dynamics in other anterior teeth, we performed H&E staining on the third deciduous incisor from E50 to E100 and found similar processes of deciduous tooth eruption and permanent tooth initiation (Appendix Fig S1).

Next, with *in situ* hybridization (ISH), we found that *Sox2*, the epithelial stem cell marker, was expressed at the tip of the SDL at E60, marking the potential location of stem cells during the resting stage (Fig 1H). We also analyzed the expression patterns of other molecules including the epithelial markers *Shh* and *Pitx2* and the mesenchymal marker *Pax9* (Fig EV1C–E). We found that *Shh* expression was absent in both the primary and successional dental lamina (Fig EV1C and Appendix Fig S2). The proliferation of dental epithelium increased significantly when the tooth bud grew into the bud stage at E90 (Fig 1I and J). However, apoptosis of dental epithelium cells remained at low levels throughout the initiation stage (Fig 1K).

In short, the SDL of the PC remained stationary after detachment from the DC germ and did not enter the bud stage until the DC erupted. The attached SDL, detached SDL, bud stage, cap stage, and bell stage could all be identified during PC development (Fig 1L).

## Difference in growth rate between the deciduous canine tooth and the alveolar socket

During the PC initiation process, we observed rapid growth of the DC. To confirm that the growth rate of the DC differed from that of the surrounding alveolar socket, we made 3-dimensional reconstructions of the DC, PC, and alveolar socket at E60 and E90 based on H&E staining of serial frontal sections (Fig 2A and Appendix Fig S3). The width of the DC increased much more rapidly than that of the labial and lingual alveolar socket (Fig 2A and B). In addition, DC width relative to total alveolar socket width increased significantly (Fig 2C). Thus, the DC width growth rate was significantly higher than that of the alveolar socket before DC eruption.

## Biomechanical stress is generated inside the mandible

Differential growth rates regulate organ development via mechanical stress or mechanical stress-linked molecular signals, indicating the existence of biomechanical stress inside tissues (Mao *et al*, 2013; Dowling *et al*, 2016; Pan *et al*, 2016; Hosseini *et al*, 2017; Qi *et al*, 2017). We therefore considered that mechanical stress may be present inside the mandible and that such mechanical stress can regulate PC initiation.

We designed an experiment to determine the level of mechanical stress inside the mandible of a miniature pig prior to DC eruption. First, we made a cut to the gingiva overlying the DC of a fresh embryonic mandible to release any possible mechanical stress. Then, deformation of the mandible caused by this release of stress was measured using micro-CT (Fig 2D–E). Finally, the quantity of stress was determined based on the deformation and mechanical features of the mandible by establishing a "cup" model using the finite element analysis software (ANSYS) (Fig 2F–J). Briefly, the input mechanical variables included mechanical properties—represented by Young's modulus from measurements and Poisson's ratio from the literature (Korhonen *et al*, 2002; Tomkoria *et al*, 2004)—and different pressures exerted on the inner wall of the mandible (Fig 2G), starting from the preset initial values. The model output was the corresponding mandible deformation. These computed deformations were compared with those from measurements to estimate the pressure (Figs 2D–J and EV2, Appendix Figs S4 and S5, and Appendix Supplementary Methods).

To measure deformation after the release of stress, micro-CT images of the E60 mandible before and after surgery were obtained and aligned to generate a 3-dimensional color map indicating the extent of deformation (Fig 2D and E). The mean ($\pm$ SEM) inward displacement of the outside compact bone was $36.48 \pm 4.04$ μm and that of the inside spongy bone was $79.74 \pm 6.07$ μm. In the control group, the size of the mandible slices before and after sham surgery was nearly identical, indicating almost no deformation (Fig 2E).

To estimate the range of stress inside the mandible at E60, we used ANSYS 15.0 to establish the cup model, in which a "cup" mimics the U-shaped mandible slice (Fig 2F–H; Quanyu *et al*, 2017). In this model, the open ends of the cup were expanded as simulated stress was applied to the inner wall (Fig 2G and H). The 3-dimensional color spectrum indicates the degree of deformation from low to high (blue to red). The optimal mesh density of the cup for the computation was established based on density tests (Fig EV2E–G).

In our model, the mandible was set as a homogeneous and elastic continuum for simplicity, in which Young's modulus and Poisson's ratio were required to run the simulations. To obtain Young's modulus of the mandible, a microdissected mandible slice was

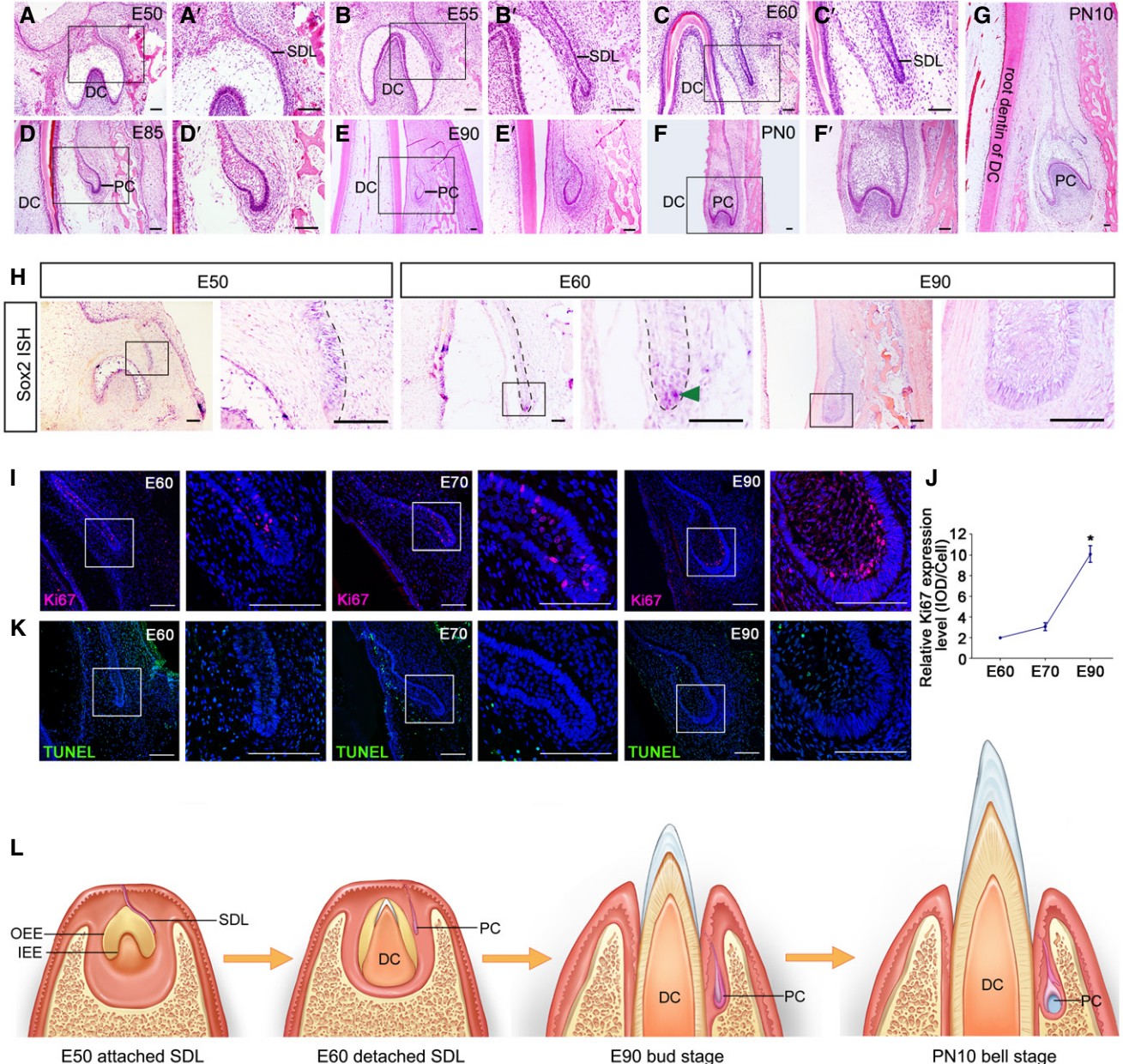

**Figure 1. Morphology and molecular map of the permanent canine germ during initiation stages in swine.**

A–G  Hematoxylin and eosin (H&E) staining of the frontal sections of miniature pig mandible slices showing morphological changes during the permanent canine (PC) initiation stage from embryonic day 50 (E50) to postnatal day 10 (PN10); (A'–F') are magnifications of boxed regions in their corresponding figure panel. DC, deciduous canine; PC, permanent canine; SDL, successional dental lamina; *n* = 3.

H  *In situ* hybridization (ISH) showing the expression pattern of *Sox2* in the initiation stage from E50 to E90; right figure panels are magnifications of boxed regions in left panels. Dashed lines mark the position of the successional dental lamina (SDL); green arrowhead indicates positive staining of *Sox2* at the tip of the SDL. *n* = 3.

I  Immunofluorescence (IF) of Ki67 at the SDL at E60, E70, and E90; right figure panels are magnifications of boxed regions in left panels. *n* = 3.

J  Semi-quantification and comparison of Ki67 expression levels during E60, E70, and E90 stages. *n* = 3.

K  TUNEL assay showing the apoptosis status of the SDL at E60, E70, and E90; right figure panels are magnifications of boxed regions in left panels. *n* = 3.

L  Diagram illustrating the initiation stage at E50 (attached SDL), E60 (detached SDL), E90 (bud stage), and PN10 (bell stage). OEE, outer enamel epithelium; IEE, inner enamel epithelium.

Data information: Data represent the means ± SEM. *$P$ < 0.05 (one-way ANOVA and Newman–Keuls post hoc tests). Scale bars = 100 μm.

chopped into small pieces (Figs 2F and EV2A) on which Young's modulus was measured using a nanoindenter (Fig EV2B and C). It was found that Young's modulus ranged from 0.1 to 0.6 MPa, with

a peak value of 0.23 MPa (Fig EV2D). As it is difficult to measure Poisson's ratio of such small samples experimentally, the impact of different Poisson's ratios on mandible deformation was evaluated

via simulation tests and the results showed small changes in mandible deformation under different ratios (0.15, 0.35, and 0.48 were tested) (Fig EV2F'); Poisson's ratio of 0.35 was ultimately used for all simulations (Appendix Supplementary Methods). The 0.35 value is the medium value of cartilage Poisson's ratio (0.15–0.45)

(Korhonen *et al*, 2002) and is within the common range in similar simulations (Tomkoria *et al*, 2004).

Deformation of the mandible was simulated when forces and two mechanical parameters (Young's modulus and Poisson's ratio) were included in the model. Based on the maximum deformation of the

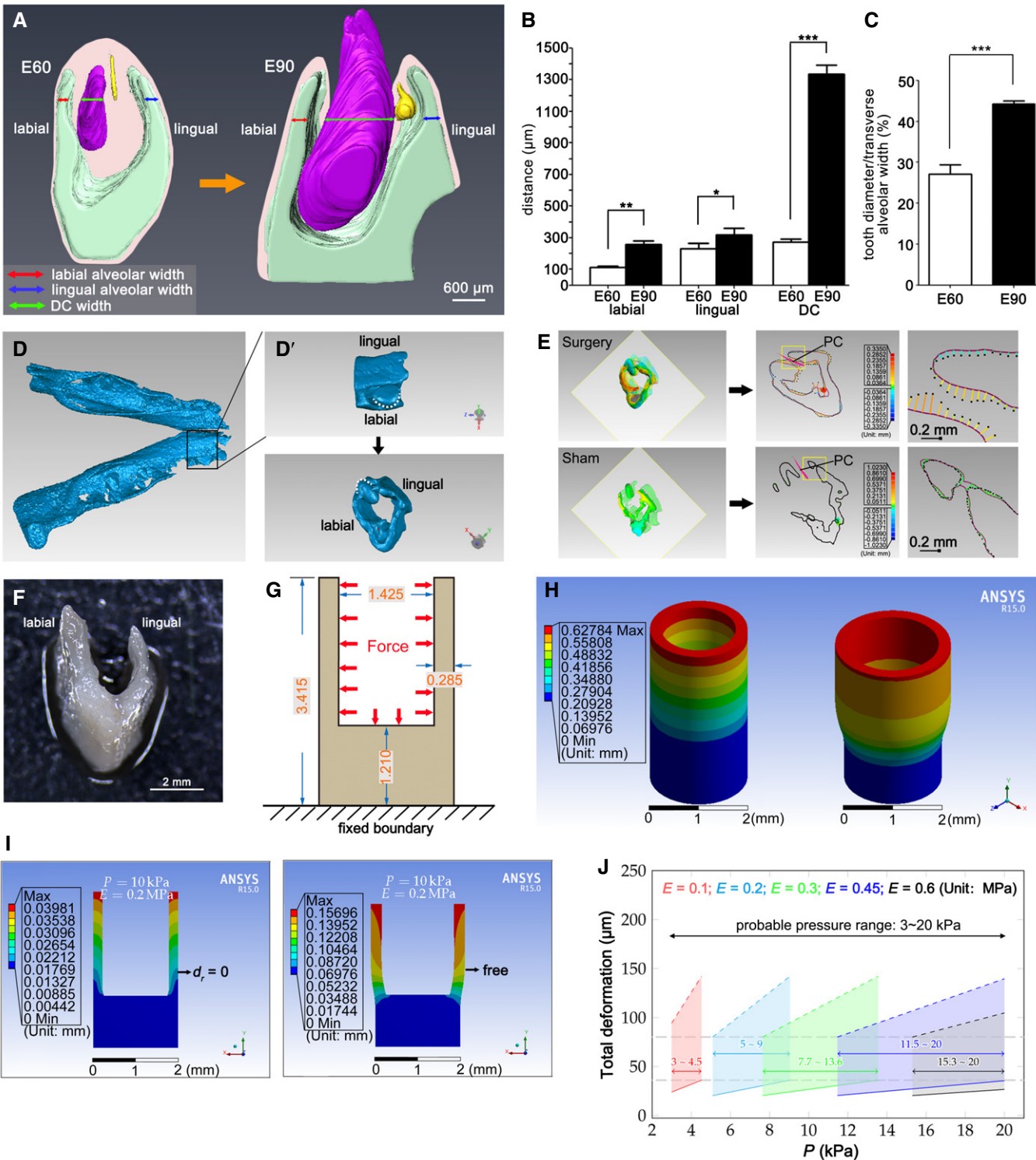

**Figure 2.**

**Figure 2. Differential growth rates of deciduous canine (DC) and alveolar socket and mechanical stress inside the mandible.**

A   Three-dimensional reconstruction of serial H&E frontal sections of miniature pig mandibles at embryonic day 60 (E60) and day E90; deciduous canine (DC) in purple, permanent canine (PC) in yellow, and alveolar socket in green. The red, blue, and green arrows indicate the width of the labial alveolar socket, lingual alveolar socket, and DC, respectively. *n* = 3.

B   DC and labial and lingual alveolar socket widths in the horizontal plane at the bottom of the PC during E60 and E90. *n* = 3.

C   The proportion of DC width relative to the total alveolar socket width during E60 and E90. *n* = 3.

D   Micro-CT imaging of the whole mandible at E60; boxed region is magnified in D' (top panel). (D') Mandible slice isolated with Geomagic software. White dashed lines mark DC.

E   3-D color map (left) after alignment of mandible slices before and after (sham) surgery showing a comparison of the surface points. Coronal sections through cusp tips (transparent squares, left) were selected for 2-D comparisons (middle); right panels are magnifications of yellow-boxed regions. The solid purple contour and dotted black contour indicate the pre- and post-surgery shapes, respectively. The distance between the two contours is the colored line segments showing the distance and direction of the movement. The colored ball in the 2-D comparison marks the position of the maximum displacement. The PC position is indicated in pink. *n* = 3.

F   Dissected mandible slice with a "U" shape.

G   In the cup model, the mandible slice was simplified as a cup according to its dimensions (unit, mm). Red arrows indicate the uniform force (stress) exerted on the inner wall of the mandible. The bottom of the cup was fixed to avoid rigid body motions.

H   3-D color map shows the extent of deformation based on the cup model established with ANSYS software.

I   Probable range of mechanical stress inside the mandible evaluated by multiple simulation tests in which serial stress and Young's modulus were inputted into the cup model; the cup was set with the outer surface fixed (left, $d_r$ = 0 denotes that radial deformation of the outer surface equals 0) or the outer surface free (right).

J   Deformation of the mandible walls under a series of stress levels with different Young's moduli. Data were obtained from multiple tests as in (I). Gray horizontal lines indicate upper and lower limit values of mean mandible wall displacement (79.74 and 36.48 μm, respectively); dashed colored lines indicate results of the free outer surface; solid colored lines indicate results of the fixed outer surface (with consideration of good linearity of the simulation results, the corresponding result points of the simulation series were omitted and replaced by solid or dashed colored lines for clarity); the actual stress value should be between the two extreme boundary conditions. Results show that the probable stress level ranged from 3 to 20 kPa. *n* = 3.

Data information: Data represent the means ± SEM. Unpaired *t*-tests, *$P < 0.05$, **$P < 0.01$, ***$P < 0.001$.

mandible walls, a stress value range of 3–20 kPa was obtained for E60 mandibles after a series of simulations (Fig 2I and J). To inspect whether gradual stress changes occur during development, a series of simulations using the mean Young's modulus of E60, E65, and E75 mandibles were performed and stress value ranges of 8.2–15.1, 6.8–8.6, and 10.8–12.7 kPa were obtained for E60, E65, and E75 mandibles, respectively (Fig EV2B–D, Appendix Figs S4 and S5, and Appendix Supplementary Methods). In conclusion, mechanical stress inside the mandible was maintained from E60 to E90 prior to DC eruption.

**Biomechanical stress determines the timing of PC initiation**

To investigate the effect of biomechanical stress on PC initiation, an E60 mandible slice containing the DC and PC was dissected and cultured. The mandible slice was obtained after cutting at the planes mesial and distal to the DC; the top gingiva of the DC was not cut (Fig 3A and Appendix Fig S6A–D). The stress inside the mandible was released when the dissection was performed. To compensate for the loss of stress, mechanical compression (3 kPa) was applied using the Flexcell FX-5000 Compression System (Fig 3A). After culturing for 2 days, the PC in the stress-free group (0-kPa control) entered the early cap stage (*n* = 12/12; Fig 3B and C). However, in the 3-kPa group, PC initiation was inhibited by the additional mechanical compression administered over 2 days (*n* = 4/8; Fig 3D). PC morphology in the 3-kPa group after 2 days resembled that of the E60 mandibles. To study whether cutting the top gingiva influences the results, we compressed the mandible slices with or without top gingiva cuts *in vitro*. The results showed that the PC was inhibited upon 3-kPa compression in both the top gingiva cut and uncut groups (Appendix Fig S7). These findings demonstrate that the PC initiation process is accelerated in the culture without stress; after only 2 days, the PC reached the early cap stage, whereas this requires more than 40 days under normal physiological conditions.

Mechanical compression of the culture compensated for the loss of stress and effectively maintained PC resting status (Fig 3E).

To study whether mechanical compression of less than 3 kPa is sufficient to inhibit PC initiation, we also performed the experiment using 1-kPa mechanical compression. However, this did not change the shape of the mandible slice. Furthermore, the success rate of the PC-inhibition phenotype was low (*n* = 1/3; Fig EV3). We also tested the effects of 10-kPa and 20-kPa compression, which produced similar results to those of the 3-kPa group (*n* = 2/3 for both 10 and 20 kPa); the widths of the alveolar socket were less than that of the 3-kPa group, however (Fig EV3). Collectively, these results indicate that biomechanical stress inside the mandible determines the timing of PC initiation.

**Biomechanical stress regulates the integrin β1-ERK1-RUNX2 pathway in the mesenchyme between DC and PC teeth**

To detect the underlying signaling mechanism of mechanical stress-mediated PC initiation, we characterized the gene expression of several mechanosensitive proteins, transforming growth factor β1 (*TGFB1*) (Buscemi *et al*, 2011), nuclear factor κB1 (*NFKB1*) (Inoh *et al*, 2002), early growth response protein 1 (*EGR1*) (Gaut *et al*, 2016), and Runt-related transcription factor 2 (*RUNX2*) (Ziros *et al*, 2002). Quantitative reverse transcription (RT–q) PCR of RNA extracted from PC tissue and surrounding connective tissue (Appendix Fig S6E–H) showed significantly higher expression levels of both *RUNX2* and *TGFB1* in the 3-kPa and E60 groups than in the 0-kPa group; however, the level of *RUNX2* expression was similar to that at E60 (Fig 3F, left panel).

We then looked at RUNX2-related mechanoreceptors on the cell surface, which can transfer biomechanical signals from the extracellular matrix to the cell and act as upstream partners of RUNX2 (Sun *et al*, 2016). We performed RT–qPCR using the following mechanoreceptors, integrins β1, β3, αV, and α2. Expression levels of

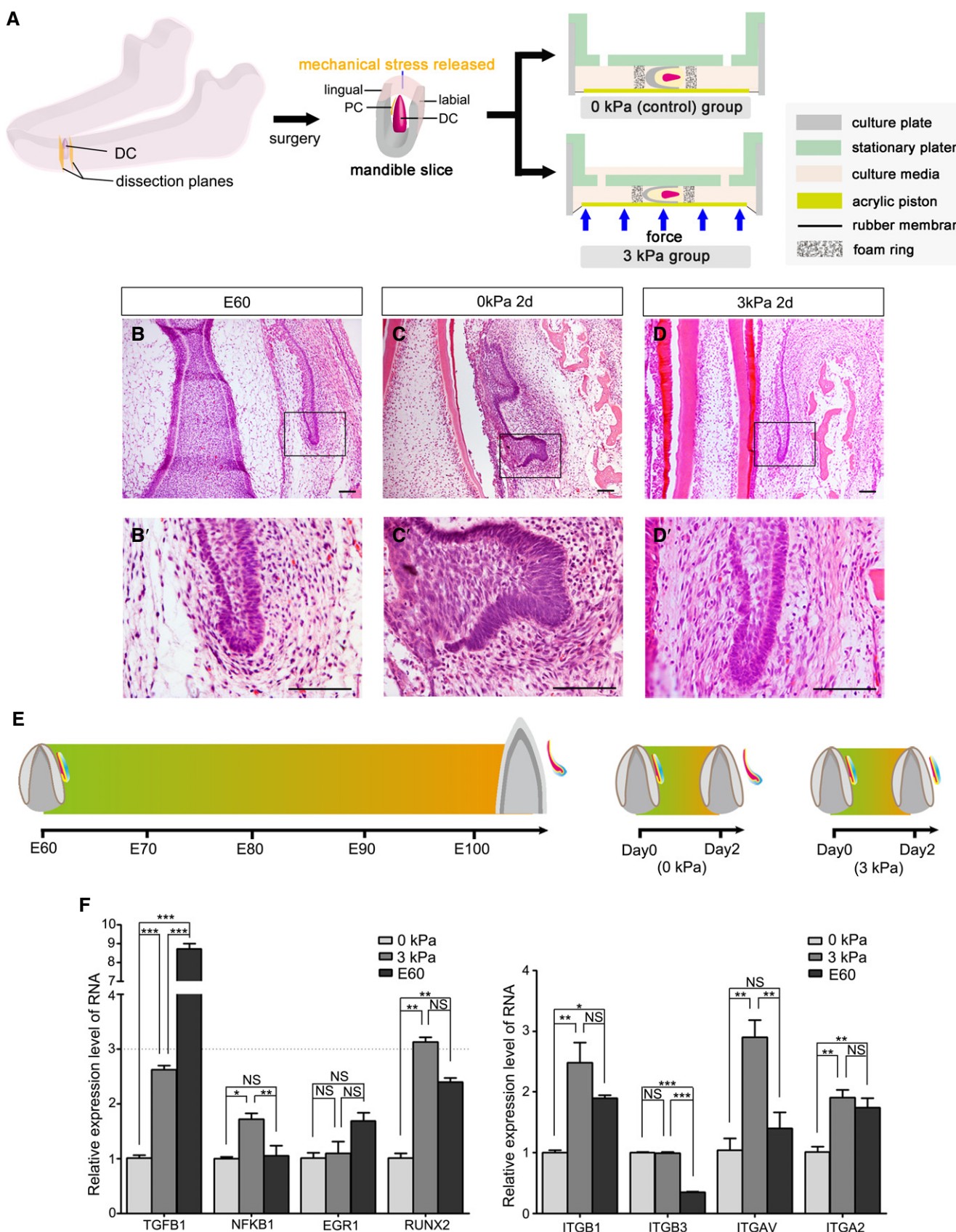

Figure 3.

**Figure 3.  Biomechanical stress determines initiation timing of PC development *in vitro*.**

A    Diagram of the dissected miniature pig mandible slice illustrating the deciduous (DC) and permanent (PC) canine and application of compression *in vitro* using Flexcell FX-5000 Compression System.

B–D  H&E staining of canine frontal sections from embryonic day 60 (E60) (B), after culturing for 2 days without stress (0 kPa) (C), or with stress (3 kPa) (D). (B'–D') are magnifications of boxed regions in their corresponding figure panels. Scale bars = 50 μm. *n* = 6/6 in E60 group; *n* = 12/12 in 0-kPa group; *n* = 4/8 in 3-kPa group.

E    Diagram illustrating the required time during PC initiation; regular process from resting to initiation stage (left) and when the mandible is cultured without stress (0 kPa; middle) or with stress (3 kPa; right).

F    RT–qPCR of several mechanosensitive proteins *(TGFB1, NFKB1, EGR1,* and *RUNX2)* and of four mechanoreceptors *(ITGB1, ITGB3, ITGAV,* and *ITGA2)* in PC and surrounding soft tissue at E60 and after culturing for 2 days with (3 kPa) or without extra mechanical stress (0 kPa). *n* = 3 for each group. Data represent the means ± SEM. One-way ANOVA (Newman–Keuls test for post hoc comparisons between two groups), *$P < 0.05$, **$P < 0.01$, ***$P < 0.001$; NS, not significant.

*ITGB1, ITGAV,* and *ITGA2* were significantly elevated in the 3-kPa group compared with the control group (0 kPa stress; Fig 3F, right panel).

The integrin β1-ERK1-RUNX2 pathway regulates osteoblast differentiation and skeletal development (Ge *et al*, 2007; Oh *et al*, 2017). Binding to integrin β1 activates the extracellular signal-regulated kinase 1 (ERK1) signaling pathway (Schlaepfer *et al*, 1994), after which ERK1 promotes *RUNX2* transcription and phosphorylation (Ren *et al*, 2015). Given that *RUNX2* and *ITGB1* expression levels were elevated in the 3-kPa group, we asked whether this pathway also plays critical roles in mechanical stress-mediated organ initiation. We used immunofluorescence (IF) to investigate the expression patterns of integrin β1, ERK1, and RUNX2. All three molecules were found expressed in the mesenchyme between the DC and PC (DC-PC mesenchyme) in the E60 pig mandible (Fig 4A–C). At E90, when stress in the mandible was released, integrin β1, ERK1, and RUNX2 were weakly expressed in the DC-PC mesenchyme (Fig 4D–F).

Similar to the E60 mandible, integrin β1, ERK1, and RUNX2 were expressed in the same region of the cultured mandible that underwent 3-kPa compression (Fig 4G–I). Similar to the E90 mandible, all three molecules were weakly expressed in the DC-PC mesenchyme of the cultured mandible without compression (Fig 4J–L). Thus, biomechanical stress helped maintain the expression levels of integrin β1, ERK1, and RUNX2 in the DC-PC mesenchyme while preserving the resting status of the PC both *in vivo* and *in vitro*.

Semi-quantitative IF analyses also confirmed that biomechanical stress helped maintain the expression levels of integrin β1, ERK1, and RUNX2 in the DC-PC mesenchyme (Fig 4M). To determine whether the RUNX2 expression pattern was similar in other permanent teeth, we performed immunohistochemistry (IHC) and ISH on the third deciduous incisor at E60 (resting stage). We observed positive signals in the mesenchyme between the DT and PT (Fig EV4), which resembled the pattern for the PC.

We also studied the expression pattern of integrin β1, ERK1, and RUNX2 in human canine tooth germs at weeks 18–19 of embryonic development. Similar expression patterns to those in pig were observed in the human DC-PC mesenchyme (Fig 4N–R).

To study the biological significance of the expression dynamics of integrin β1, ERK1, and RUNX2, we evaluated the phenotype of PC initiation by quantifying the extent of epithelial thickening. By measuring the area of the PC epithelium, we found that it was significantly larger both at E90 than at E60 and in the 0-kPa group than in the 3-kPa group (Fig 4S).

Taken together, these results indicate that the biomechanical stress-associated integrin β1-ERK1-RUNX2 pathway in the DC-PC mesenchyme may play critical roles in regulating PT initiation.

## Mechanical force regulates RUNX2 via the integrin β1-ERK1-RUNX2 pathway

We next examined whether RUNX2 expression can be regulated by exerting force on cells via the integrin β1-ERK1-RUNX2 pathway. Given that the DC-PC mesenchyme is located outside the outer enamel epithelium and is occupied by the dental follicle of the DC (Wise & Yao, 2006), we separated the dental follicle tissue of the DC and cultured the primary dental follicle cells (DFCs).

Static compression was applied to DFCs using the weighted glass coverslip method (Fig 4T; Feng *et al*, 2016). To determine the optimum force value and loading time, we first analyzed the relative IF expression levels of RUNX2 and p-ERK1/2 upon force loading with $1.0 \, g/cm^2$ for 0, 1, 2, and 4 h and found that expression levels were highest at 2 h (Appendix Fig S8A–D). To determine the optimum force value, we analyzed the relative IF expression levels upon force loading with 1.0, 2.0, and $5.0 \, g/cm^2$ for 2 h and found that expression levels were highest at $1.0 \, g/cm^2$ (Appendix Fig S8E–H). To study whether activation of RUNX2 and p-ERK1/2 can be sustained or is reversible after the force is removed, we compressed the cells with $1.0 \, g/cm^2$ for 2 h, then removed the force, and cultured for another 1, 2, or 4 h before harvesting the cells. After analyzing the IF expression levels, we found that RUNX2 expression levels can be sustained for 2 h, while those of p-ERK1/2 can be sustained for only 1 h after removing the force. Activation was downregulated for both 4 h after removing the force (Fig 4U–V). These results indicate that RUNX2 and p-ERK1/2 activation is reversible when the force is removed on the cellular level.

Western blotting also showed upregulated expression of phospho-ERK1/2 (p-ERK1/2) and RUNX2 in DFCs upon loading the cells with $1.0 \, g/cm^2$ for 2 h (Fig 4W and X). These findings provided direct evidence that an external force regulates p-ERK1/2 and RUNX2 expression levels in a time- and force-dependent manner.

While the cells were compressed, we also added an anti-integrin β1 neutralizing antibody (5 μg/ml) to the culture medium for 2 h. This resulted in significantly lower p-ERK1/2 and RUNX2 expression levels (Fig 4W and X), indicating that mechanotransduction from the extracellular matrix to the DFCs is mediated by integrin β1. To investigate whether ERK1 is the mechanotransducer from integrin β1 to RUNX2, we treated compressed cells with U0126, an inhibitor of ERK1 (Ren *et al*, 2015). After treatment for 2 h, p-ERK1/2 and RUNX2 expression levels were reduced (Fig 4W and X), indicating that mechanical compression upregulates RUNX2 expression in DFCs via the integrin β1-ERK1-RUNX2 pathway.

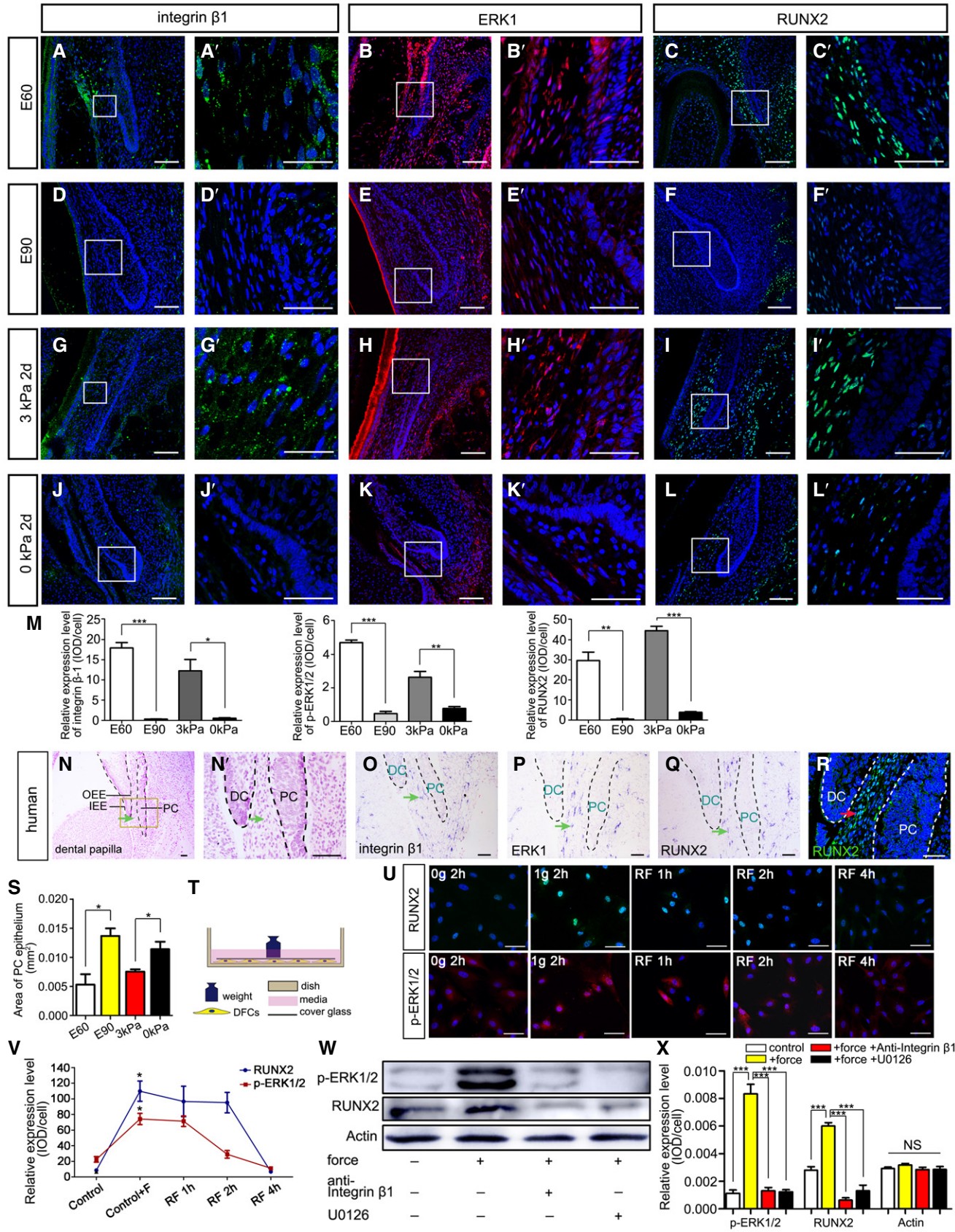

**Figure 4.**

**Figure 4.  Biomechanical stress regulates the integrin β1-ERK1-RUNX2 pathway in mesenchyme between DC and PC.**

A–L    IF of integrin β1, ERK1, and RUNX2 in miniature pig canine frontal sections at embryonic days 60 (E60), E90, and E60 cultured under 3 kPa stress for 2 days and E60 cultured under 0 kPa stress for 2 days; (A'–L') are magnifications of boxed regions in the corresponding figure panels.

M      Relative IF expression levels of integrin β1, ERK1, and RUNX2 during E60, E90, and E60 cultured under 3 kPa stress for 2 days and E60 cultured under 0 kPa stress for 2 days.

N      Morphology of the human PC at weeks 18–19 (H&E staining); (N') is magnification of the boxed region in (N). DC, deciduous canine; PC, permanent canine; IEE, inner enamel epithelium; OEE, outer enamel epithelium. Mesenchyme between DC and PC is indicated with a green arrow.

O–Q    ISH expression patterns of *ITGB1*, *ERK1*, and *RUNX2* in human PCs. Green arrows indicate the mesenchyme between DC and PC.

R      IF of RUNX2 in human PC. Mesenchyme between DC and PC is indicated with a red arrow. Red arrow indicates the mesenchyme between DC and PC.

S      PC epithelium areas in E60, E90, and E60 cultured under 3 kPa stress for 2 days and E60 cultured under 0 kPa stress for 2 days.

T      Illustration of force exertion on cultured dental follicle cells (DFCs).

U      IF of RUNX2 and phospho-ERK1/2 (p-ERK1/2) upon force loading with 0 or 1.0 g/cm$^2$ for 2 h or after force was removed and the cells cultured for an additional 1, 2, and 4 h.

V      Relative IF expression levels of RUNX2 and p-ERK1/2 between groups in (U).

W      Western blots of p-ERK1/2 and RUNX2 levels after 1.0 g/cm$^2$ was applied for 2 h, with or without anti-integrin β1 antibody and ERK1 inhibitor (U0126).

X      Relative expression levels of p-ERK1/2, RUNX2, and Actin in (W).

Data information: Data represent the means ± SEM. $n = 3$. Scale bars = 25 μm (A', G') and 50 μm (other panels). Unpaired *t*-tests for (M); one-way ANOVA (Newman–Keuls test for post hoc comparisons between two groups) for( V and X), *$P < 0.05$, **$P < 0.01$, ***$P < 0.001$; NS, not significant.

Source data are available online for this figure.

### RUNX2 overexpression inhibits PC initiation

We next investigated the functional role of RUNX2 in PC initiation. First, a lentiviral vector encoding *RUNX2* (LV-*RUNX2*) was constructed to determine whether *RUNX2* overexpression inhibits PC initiation in the mandible *in vitro*. PC initiation was inhibited 2 days after transfection of the explanted mandible slice without a gingiva cut with LV-*RUNX2*, resembling embryonic development at E60 (Fig 5A and B). Conversely, in the overexpression control lentiviral vector (O-control) group, the PC developed into the early cap stage (Fig 5C). IHC to detect the Myc tag confirmed successful transductions (Fig 5D–F). IF showed minimal RUNX2 expression in the DC-PC mesenchyme in the O-control group, in contrast with the overexpression and E60 groups (Fig 5G–I). The immunostaining intensities for Myc and RUNX2 were semi-quantified and supported the above observations (Fig 5J and K).

We also conducted *RUNX2* knockdown experiments. After infection with a *RUNX2* short-hairpin RNA (shRNA) lentiviral vector (knockdown), RUNX2 expression was downregulated compared with that of the scrambled shRNA lentiviral vector (knockdown control, K-control) group. The PC grew to the early cap stage, which was similar to the PC in the K-control group (Fig 5L and M). Semi-quantification of IF intensities showed a significant reduction in RUNX2 expression in the knockdown group (Fig 5N).

Next, we investigated whether additional mechanical compression can reverse the phenotype induced by the *RUNX2* shRNA lentiviral vector. We applied 3-kPa compression to mandible tissue that was infected with the *RUNX2* shRNA lentiviral vector (knockdown + force group) using the method described in Fig 3. The PC still grew to the cap stage, regardless of compression (Fig 5O and P). However, in the K-control + force group, PC initiation was inhibited by the 3-kPa compression applied (Fig 5Q). In addition, RUNX2 expression was significantly decreased in the K-control and knockdown + force groups compared with the K-control + force group (Fig 5R–U).

To confirm and evaluate the phenotype of PC initiation or inhibition with RUNX2 overexpression or knockdown, we quantified the extent of epithelial thickening by measuring the PC epithelium area. The results confirmed the phenotypes observed above (Fig 5V). Thus, *RUNX2* overexpression restrains PC initiation.

### Wnt modulation between the mesenchyme and epithelium regulates PC initiation

Wnt signaling plays critical roles in the initiation of tooth germs in mice (Jarvinen *et al*, 2006; Liu *et al*, 2008). Activation of Wnt signaling has also been observed in the SDL of reptiles (Handrigan & Richman, 2010) and β-catenin—an intracellular mediator of canonical Wnt signaling (Liu *et al*, 2008)—may be important in PT initiation in humans. In pig tooth germs, expression patterns of *RUNX2* and *β-catenin* were colocalized in the DC-PC mesenchyme at E60 and E90 (Fig 6A, B, D, and E). Semi-quantification of *RUNX2* and *β-catenin* expression levels showed a synchronous decrease in the mesenchyme from E60 to E90 (Fig 6C and F). β-catenin and Lef1 were expressed in both the mesenchyme and epithelium at E60 and were mainly expressed in the epithelium at E90 (Fig 6D–I). This means that Wnt signaling was downregulated in the DC-PC mesenchyme and activated in the tip of the enamel organ during PC initiation.

We also investigated the expression patterns of innate Wnt inhibitor molecules *Sfrp1* and *Sostdc1* (Kawano & Kypta, 2003; Prochazkova *et al*, 2017). *Sfrp1* was expressed on the outer layer of the SDL of the PC (Fig EV5A), while *Sostdc1* was expressed on the inner layer from E50 to E90 (Fig EV5B). This indicates that Wnt inhibitors are also expressed in the epithelium during the PT initiation process.

Given that Wnt signaling was enhanced in the tip of the enamel organ at E90, we investigated whether the PC can be initiated by direct activation of Wnt signaling. We added a Wnt pathway stimulator (LiCl) or an inhibitor (recombinant DKK1 protein) to the culture medium of E60 mandible slices for 2 days. PC initiation occurred in both the LiCl and control groups. In contrast, the dental lamina remained stationary in the DKK1 group (Fig EV5C–H).

To study the dynamics of Wnt signaling both in the mesenchyme and in the epithelium, we cultured E60 mandible slices for 12 and 24 h and compared *β-catenin* expression levels between the nucleus

and cytoplasm among the E60 (cultured for 0 h), E90, E60 (cultured for 12 h), and E60 (cultured for 24 h) groups via an RNAscope assay—an RNA ISH technique for single-molecule detection that can identify the subcellular location of signal (i.e., in the nucleus and cytoplasm) (Arneson *et al*, 2018; Schulz *et al*, 2018). Quantification results indicated that during PC initiation from E60 to E90, the nuclear *β-catenin* signal of the mesenchyme decreased approximately 10-fold, while the nuclear *β-catenin* signal of the epithelium increased approximately threefold. In the mandible explant *in vitro*, a similar tendency in both the epithelium and mesenchyme was observed after 12 h in culture (Fig 6J and K). At the protein level, the nuclear β-catenin in the mesenchyme decreased, while that in the epithelium increased from E60 to E90, which was similar to RNA level (Fig 6L–N). Thus, these results support the notion that Wnt signals are relayed from the mesenchyme to the epithelium during PC initiation.

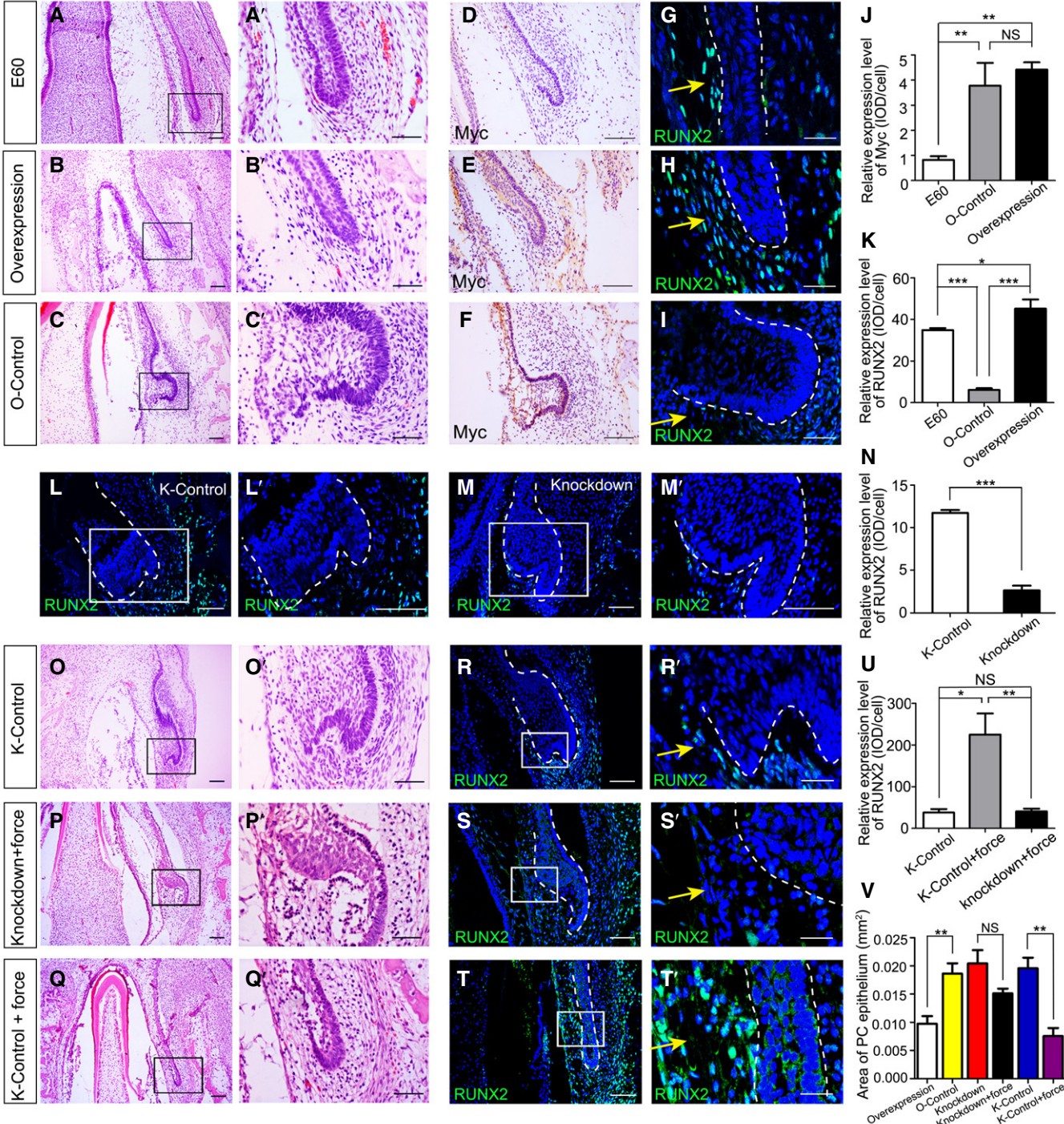

**Figure 5.**

◀

**Figure 5.  *RUNX2* overexpression inhibits initiation of PC germs.**

A–C    H&E staining of miniature pig canine frontal sections from embryonic day 60 (E60 group) (A) and after the mandible slices were infected with *RUNX2* overexpression lentiviral vector with a Myc tag (overexpression group) (B) or overexpression control lentiviral vector with a Myc tag (O-control group) (C); (A'–C') are magnifications of boxed regions in the corresponding figure panels.

D–F    Immunohistochemical (IHC) staining of Myc expression in mandible slices from the E60, overexpression, and O-control groups.

G–I    IF staining of RUNX2 expression (yellow arrows) in mandible slices of the E60, overexpression, and O-control groups.

J, K    Relative expression levels of Myc and RUNX2 in the E60, O-control, and overexpression groups.

L, M    IF staining of RUNX2 expression after infecting mandible slices with scrambled shRNA lentiviral vector (knockdown control, K-control) (L) or *RUNX2* shRNA lentiviral vector (knockdown) (M); (L'–M') are magnifications of boxed regions in the corresponding figure panels.

N    Relative RUNX2 expression levels between K-control and knockdown groups.

O–Q    H&E staining of mandible slice sections from the K-control, knockdown + force (3 kPa), and K-control + force (3 kPa) groups. (O'–Q') are magnifications of boxed regions in the corresponding figure panels.

R–T    IF staining of RUNX2 expression (yellow arrows) of mandible slice sections from the K-control, knockdown + force (3 kPa), and K-control + force (3 kPa) groups. (R'–T') are magnifications of boxed regions in the corresponding figure panels.

U    Relative RUNX2 expression levels among the three groups in (R–T).

V    PC epithelium areas in the overexpression, O-control, knockdown, knockdown + force, K-control, and K-control + force groups.

Data information: Data represent the means ± SEM. *n* = 3 for all experiments. Scale bars = 100 μm (D–F), 25 μm (R'–T'), and 50 μm (other panels). Dashed lines mark the outlines of PCs. Unpaired *t*-tests for (N and V); one-way ANOVA (Newman–Keuls test for post hoc comparisons between two groups) for (J, K, and U), *$P < 0.05$, **$P < 0.01$, ***$P < 0.001$.

Given that *RUNX2* was colocalized with *β-catenin* in the DC-PC mesenchyme, we asked whether a reduction in RUNX2 activity in the mesenchyme would lead to activation of Wnt signaling in the epithelium. After analyzing the signal in the nucleus vs. the cytoplasm via RNAscope, we found that expression levels of both nuclear and cytoplasmic *β-catenin* were elevated significantly in the samples infected with *RUNX2* shRNA lentivirus (Fig 6O–Q). This indicates that the reduction in RUNX2 activity in the mesenchyme leads to activation of Wnt signaling in the epithelium.

**RUNX2 regulates Wnt signaling in the DC-PC mesenchyme**

To study whether RUNX2 acts upstream of Wnt signaling in the mesenchyme between DC and PC, we infected samples with *RUNX2* overexpression lentivirus and found that Lef1 expression was maintained in both the epithelium and mesenchyme (Fig 7A–C). Similarly, Lef1 expression was maintained in both the epithelium and mesenchyme in mandible slices subjected to 3-kPa compression for 2 days (Fig 7D–F).

Next, we studied whether RUNX2 acts upstream of Wnt signaling in cultured DFCs. The expression levels of Wnt signaling molecules were examined by Western blot analysis after DFCs were infected with a *RUNX2* overexpression lentiviral vector or a *RUNX2* shRNA (knockdown) lentiviral vector. Lef1 and β-catenin expression was elevated in the *RUNX2* overexpression group (Fig 7G) but decreased in the *RUNX2* knockdown group (Fig 7H). Translocation of β-catenin from the cytoplasm to the nucleus indicates activation of Wnt signaling (Kikuchi *et al*, 2006); therefore, we analyzed nuclear β-catenin expression and found it upregulated in the *RUNX2* overexpression group and downregulated in the *RUNX2* knockdown group compared with control (Fig 7I, left and middle panels). Similarly, nuclear β-catenin was upregulated in the compressed DFCs compared with the control group (Fig 7I, right panel and Appendix Fig S9).

To further study whether the increase in nuclear β-catenin levels is due to activation of Wnt signaling or an increase in total β-catenin expression, we added IWR-1-endo—an inhibitor of Wnt signaling (Martins-Neves *et al*, 2018)—to the culture system while compressing cells or infecting them with *RUNX2* overexpression lentivirus.

The results showed that nuclear β-catenin levels were reduced in the Wnt inhibitor group (Fig 7J). Thus, the findings demonstrated that the force-mediated RUNX2-Wnt signaling pathway regulates Wnt signaling in the DC-PC mesenchyme.

## Discussion

In our miniature pig model and human canine germ samples, biomechanical stress regulated the rhythm of tooth replacement. PT development was modulated by the integrin β1-ERK1-RUNX2-Wnt/β-catenin pathway in the mesenchyme between the DT and PT (Fig 7K). Downregulation of this pathway in the mesenchyme activated Wnt signaling in the PT epithelium, triggering PT development (Fig 7L).

### Transition from resting to initiation stage in integumentary organs

The SDL can rest for a period of time without giving rise to a replacement tooth (Jernvall & Thesleff, 2012). In this study, we found that mechanical stress inside the mandible regulates the transition from resting to initiation stages of replacement teeth in large mammals. The PT did not begin transitioning from resting to initiation stage until the DT erupted. To the best of our knowledge, this is the first study to report that the mechanical stress inside the mandible regulates the initiation process of tooth replacement in large mammals. Notably, the molecular expression patterns in the resting stage of human PCs were found to be similar to those of miniature pigs.

### Critical role of mechanical stress in organ development and regeneration

Biomechanical forces participate in an organism's development (Shawky & Davidson, 2015). Mechanical heterogeneity is believed to result in hair regeneration in large excisional wounds (Davidson, 2018; Guerrero-Juarez *et al*, 2018). In patients with aplastic lower second premolars, treatment of localized lower molar mesialization

                                                                                   

results in the development and eruption of wisdom teeth (Zimmer, 2006), indicating that mechanical stress from the adjacent tooth or jaw bone inhibits the development process of teeth. On the other hand, lateral mechanical constraint from the growing mandible facilitates the formation of the specific shape of tooth crowns (Renvoise *et al*, 2017). In our study, the mechanical constraint of the mandible repressed the initiation of PT development and the release of biomechanical stress by DT eruption triggered the initiation process. Therefore, there was mechanical heterogeneity in the mandible before and after eruption of the DT.

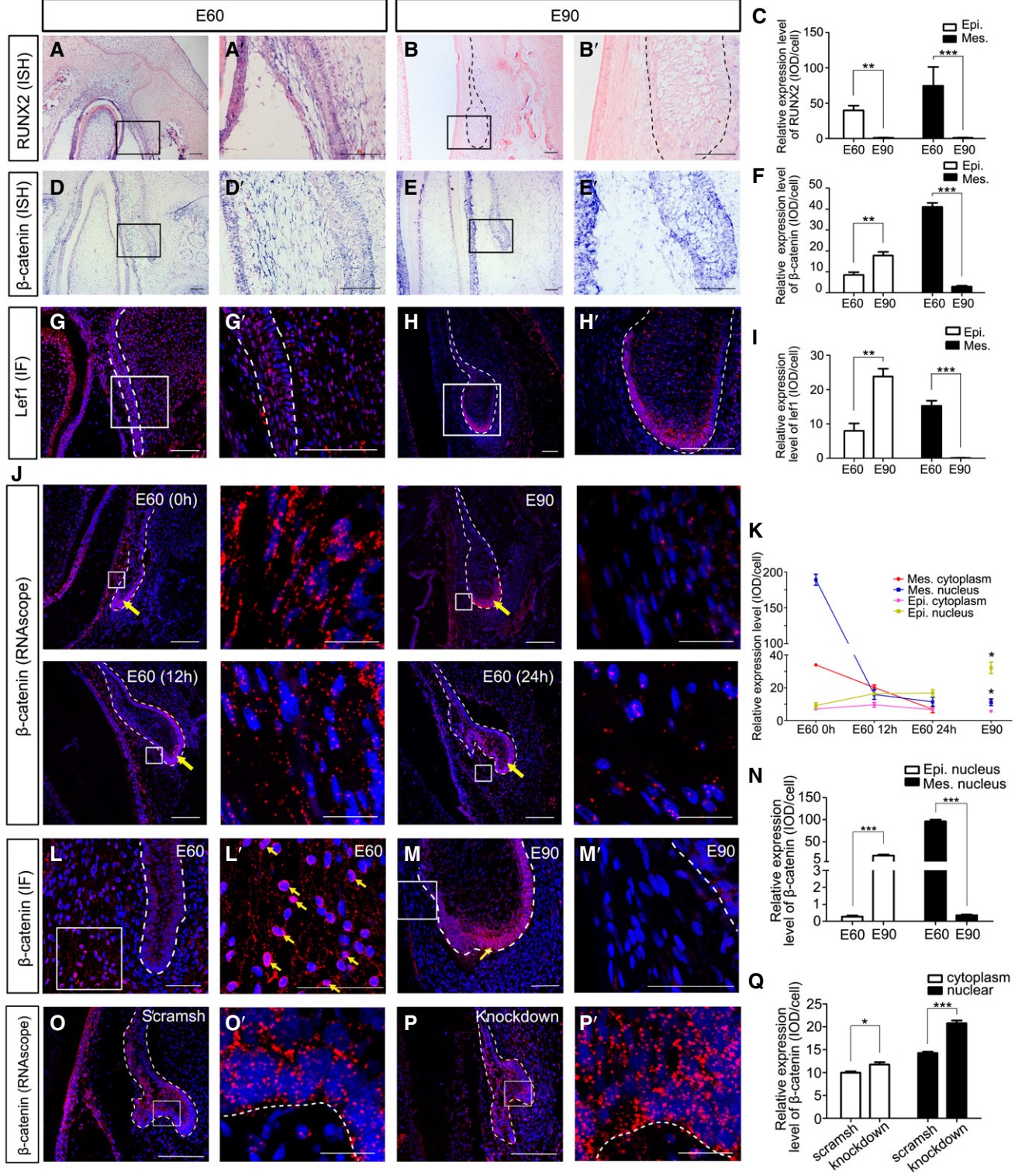

**Figure 6.**

**Figure 6. Wnt modulation between the mesenchyme and epithelium regulates PC initiation.**

A, B   ISH of miniature pig canine frontal sections showing expression patterns of *RUNX2* at embryonic days 60 (E60) and E90. (A', B') are magnifications of boxed regions in the corresponding figure panels.

C   Relative ISH expression levels of *RUNX2* in epithelium (Epi.) and mesenchyme (Mes.) at E60 and E90.

D, E   ISH of miniature pig canine frontal sections showing expression patterns of *β-catenin* at E60 and E90. (D', E') are magnifications of boxed regions in the corresponding figure panels.

F   Relative ISH expression levels of *β-catenin* in epithelium (Epi.) and mesenchyme (Mes.) at E60 and E90.

G, H   IF of Lef1 at E60 and E90. (G, H') are magnifications of boxed regions in the corresponding figure panels.

I   Relative IF expression levels of Lef1 in epithelium (Epi.) and mesenchyme (Mes.) at E60 and E90.

J   FISH of *β-catenin* in E60 and E90 mandible slices and E60 mandible slices after 12 and 24 h in culture via RNAscope. Boxed regions of mesenchyme are magnified in the right panels. Epithelium is indicated with a yellow arrow.

K   Relative FISH expression levels of *β-catenin* in the nucleus and cytoplasm of epithelium and mesenchyme at E60, E90, and E60 cultured for 12 and 24 h.

L, M   IF of β-catenin at E60 and E90; the yellow arrows indicate the localization of β-catenin in the nucleus.

N   Relative IF expression levels of β-catenin in the nucleus of epithelium and mesenchyme at E60 and E90.

O, P   FISH of *β-catenin* in mandible slices infected with *RUNX2* shRNA lentivirus (knockdown) or scrambled shRNA lentivirus (scramsh) via RNAscope.

Q   Relative FISH expression levels of *β-catenin* in the nucleus and cytoplasm of epithelium and mesenchyme in knockdown and scramsh groups.

Data information: Scale bars = 50 μm. Data represent the means ± SEM. $n = 3$ for all experiments. Unpaired *t*-tests for (C, F, I, N, and Q); one-way ANOVA for (K), *$P < 0.05$, **$P < 0.01$, ***$P < 0.001$.

To estimate the range of stress inside the mandible, we established the cup model, in which a cup mimics the U-shaped mandible slice. Our model effectively captured the mechanical behavior of the embryonic mandible and we expect it to be a good model for stress assessment in other organs and tissues.

In other studies, the loss of a functional organ, such as from hair plucking, tooth extraction, or physiological molting, stimulates the development of the next generation of the organ, which possibly also results from mechanical stress relief (Wu *et al*, 2013; Chen *et al*, 2015). In the present study, biomechanical stress caused by differential growth rates inhibited the development of the next-generation tooth. This inhibition was lifted when the stress was released. However, it is possible that initiation of the PC *in vivo* is regulated by other signals, while changes in its development in explants can be artificially influenced by force.

**Central role of RUNX2-Wnt signaling emanates from the mesenchyme/dental lamina**

A loss-of-function mutation in human *RUNX2* causes a supernumerary teeth phenotype, including excessive formation of replacement teeth and supernumerary posterior molars (Jensen & Kreiborg, 1990). In a recent study, supernumerary teeth caused by a *RUNX2* loss-of-function mutation were found to result from decreased mesenchymal Wnt signaling due to the upregulation of the Wnt inhibitors AXIN2 and DRAPC1. Also, forced activation of Wnt signaling in the dental mesenchyme inhibits the formation of next-generation teeth in mice (Jarvinen *et al*, 2018). Here, we found that *RUNX2* overexpression maintained the SDL of the PT at rest, while *RUNX2* knockdown leads to the activation of Wnt activity in the epithelium and stimulated initiation of an aberrant PT bud. These findings explain the phenotype of supernumerary teeth and are also consistent with findings in a previous study (Jarvinen *et al*, 2018). However, the mechanism of signal relay from the mesenchyme to the epithelium warrants further investigation.

**Similarities and differences between replacement and cascade addition of teeth**

There are two types of sequential tooth formation, tooth replacement and cascade addition (Jernvall & Thesleff, 2012). Tooth replacement usually occurs continuously in fish and reptiles but only once in mammals (Handrigan *et al*, 2010). The replacement tooth develops from the SDL located on the lingual side of the preceding tooth (Huysseune, 2006). The functional and replacement tooth is located in the same tooth position, with the replacement occurring vertically (Wu *et al*, 2013). However, cascade addition usually occurs in the posterior molars in mammals. The posterior molar develops from the distal end of the SDL protruding from the anterior neighboring molar (Jarvinen *et al*, 2018). In miniature pigs, cascade addition starts from the third deciduous molar and stops distally at the third permanent molar (Wang *et al*, 2017). Development from the free end of the SDL renders the mechanism of these two formation types quite similar. The epithelial stem cell marker Sox2 is found in the free end of the SDL in both types (Juuri *et al*, 2013). Although the SDL is located at the lingual side of the precedent tooth in both types, the mechanism by which the new tooth bud develops in the vertical end of the SDL to replace the precedent one or in the distal end of the SDL in cascade addition remains largely unknown.

In this work, we linked the release of mechanical stress to the initiation of PC development in a series of *in vitro* experiments. We then linked the mechanics at the tissue level to events at the molecular level. The defined biomechanical stress modulation and its impact on Wnt signaling between organ epithelium and the surrounding mesenchyme serves as an important paradigm for future studies on integumentary organ regeneration.

# Materials and Methods

### Animal studies

All animal experiment procedures were approved by the Animal Care and Use Committee of the Capital Medical University (Beijing, China; permit number: AEEI-2016-063), and the committee confirmed that all experiments conformed to the relevant regulatory standards. Pregnant miniature pigs (Chinese Wuzhishan strain) were obtained from the Institute of Animal Science of the Chinese Agriculture University (Beijing, China). The age of the embryos was based on the day of insemination. Pregnancy was verified by B-type ultrasonic inspection.

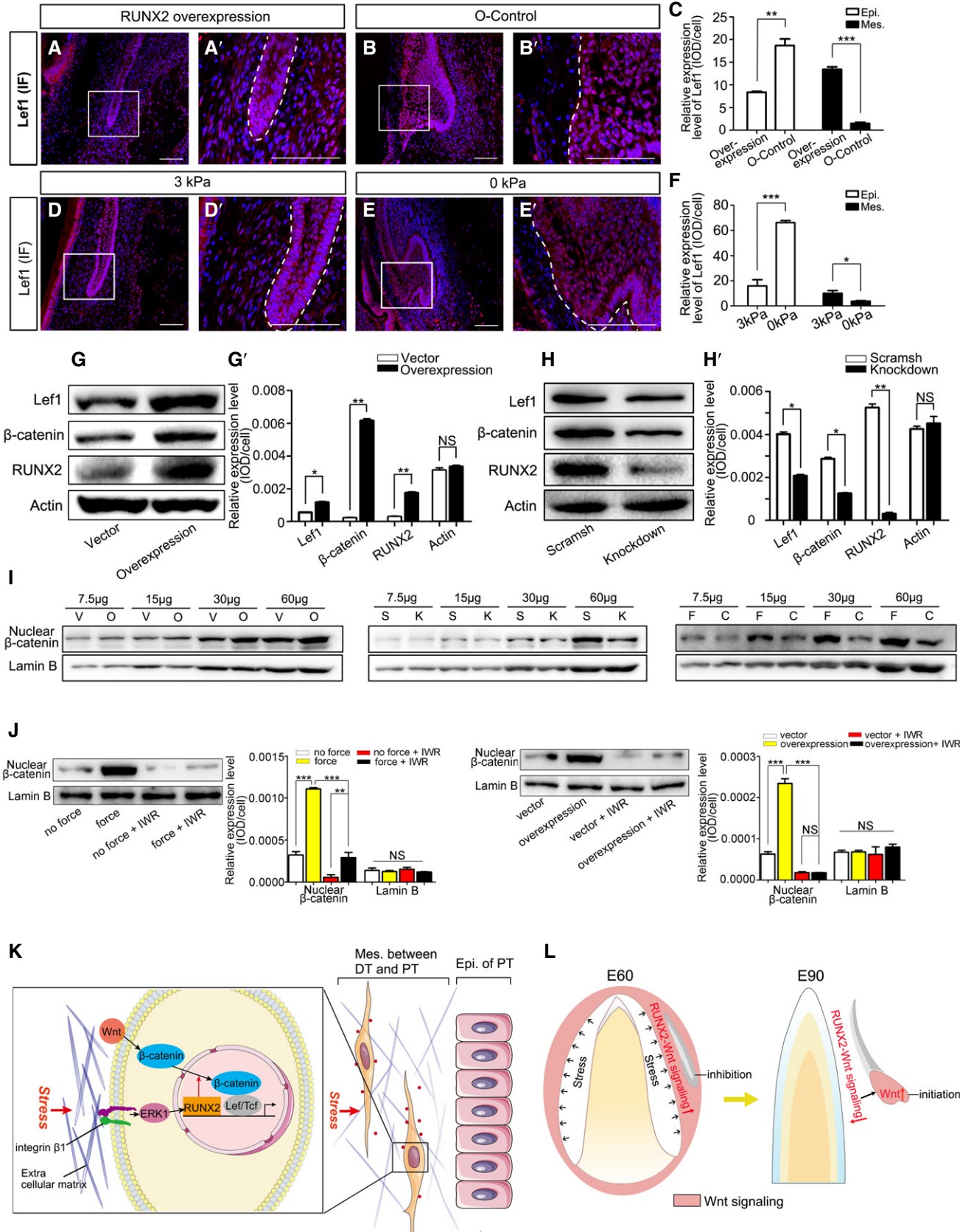

**Figure 7.**

**Figure 7. RUNX2 regulates Wnt signaling in DFCs.**

A, B   IF of Lef1 in mandible slices at embryonic day 60 (E60) infected with *RUNX2* overexpression lentiviral vector (overexpression) or overexpression control lentiviral vector (O-control). (A′–B′) are magnifications of boxed regions in the corresponding figure panels. Dashed lines mark the epithelium of PC.

C   Relative IF expression levels of Lef1 in epithelium (Epi.) and mesenchyme (Mes.) of the overexpression and O-control groups.

D, E   IF of Lef1 in E60 mandible slices subjected to 3 or 0 kPa stress for 2 days. (D′, E′) are magnifications of boxed regions in the corresponding figure panels. Dashed lines mark the epithelium of PC.

F   Relative IF expression levels of Lef1 in the epithelium and mesenchyme after applying 3 and 0 kPa pressure.

G   Western blots of Lef1, non-phospho-β-catenin, and RUNX2 after DFCs were infected with control lentiviral vector or *RUNX2* overexpression lentiviral vector. (G′) Relative expression levels between control vector and *RUNX2* overexpression groups.

H   Western blots of Lef1, non-phospho-β-catenin, and RUNX2 after DFCs were infected with scrambled shRNA (scramsh) or *RUNX2* knockdown shRNA (knockdown) lentiviral vectors. (H′) Relative expression levels between scramsh and knockdown groups.

I   Western blots of nuclear non-phospho-β-catenin and Lamin B in DFCs of *RUNX2* overexpression (O), control vector (V), *RUNX2* knockdown (K), scramsh (S), compressed force (F; 1.0 g/cm$^2$), and control (C; 0 g/cm$^2$) groups. Sample was loaded in a gradient (7.5, 15, 30, and 60 μg) showing a linear relationship for each group.

J   Western blots of nuclear non-phospho-β-catenin and Lamin B in DFCs treated with IWR-1-endo. Relative expression levels were compared between force and no force groups and between overexpression and control groups with or without IWR-1-endo treatment.

K   Diagram illustrating the biomechanical stress regulation of Wnt/β-catenin signaling in the mesenchyme between the deciduous (DT) and permanent tooth (PT) via the integrin β1-ERK1-RUNX2-Wnt/β-catenin pathway.

L   Diagram illustrating the biomechanical stress-associated downregulation of RUNX2-Wnt/β-catenin pathway in the mesenchyme, inducing upregulation of Wnt signaling in the epithelium, which triggers PT development.

Data information: Data represent the means ± SEM. Scale bars = 100 μm. $n = 3$ for all experiments. Unpaired *t*-tests for (C, F, G′, and H′); one-way ANOVA (Newman–Keuls test for post hoc comparisons between two groups) for (J), *$P < 0.05$, **$P < 0.01$, ***$P < 0.001$; NS. not significant.

Source data are available online for this figure.

## Human embryonic canine germ studies

The procedures for collecting human embryos were approved by the Yidu Central Hospital, Weifang Medical University (Weifang, China; permit number: 035). Three healthy embryos between 18 and 19 weeks of age were donated by women who underwent induced abortion. Informed consent was obtained from all subjects. The experiments conformed to the principles set out in the WMA Declaration of Helsinki and the Department of Health and Human Services Belmont Report.

## Tissue preparation for histological analysis

Mandible samples from miniature pigs and human embryos were fixed in 4% paraformaldehyde in phosphate-buffered saline (PFA-PBS; Sigma-Aldrich, St. Louis, MO) at 4°C overnight. The pig mandible slice containing a canine or third incisor as well as the human mandible slice containing a canine was dissected. After rinsing with PBS for 10 min twice, pig mandible slices from embryonic days 50 (E50), E55, E60, and E70 were decalcified in 10% EDTA-PBS for 2, 4, 7, and 14 days, respectively. Pig mandible slices from E85, E90, E100, PN0, and PN10 were decalcified in 10% EDTA-PBS for 30 days. Human mandible slices aged 18–19 weeks were decalcified in 10% EDTA-PBS for 30 days. The decalcification media were refreshed every 2 days.

Samples were dehydrated in serial alcohol dilutions (30, 50, 70, 90, 95, and 100%) and embedded in paraffin, after which the samples were sectioned (5–7 μm thickness) for staining. H&E staining was used for morphological examination. Detailed ISH, TUNEL, RNAscope, and quantification methods are provided in the Appendix Supplementary Methods.

## Immunofluorescence and histochemistry

Paraffin sections were heated, deparaffinized, and rehydrated. After antigen retrieval treatment, sections were immersed in 10% H$_2$O$_2$/methanol for 10 min to bleach and to quench the endogenous peroxidase activity. Then, sections were incubated with primary antibodies at 4°C overnight. After rinsing for 10 min three times, sections were incubated with secondary antibodies (horseradish peroxidase-conjugated or Alexa Fluor series) at room temperature for 2 h. Immunofluorescent (IF) images were taken using a Leica confocal microscope. Immunohistochemical (IHC) signals were detected with a DAB substrate kit (8059; Cell Signaling Technology, Danvers, MA).

The primary antibodies (1:200 dilution) used were as follows: mouse monoclonal anti-integrin β1 (ab30388; Abcam, Cambridge, UK); rabbit monoclonal anti-phospho-ERK1/ERK2 (p-ERK1/2) (MA5-15173; Thermo Fisher Scientific, Waltham, MA); mouse monoclonal anti-RUNX2 (sc-390351; Santa Cruz Biotechnology, Dallas, TX); mouse monoclonal anti-Myc tag (2276S; Cell Signaling Technology); rabbit polyclonal anti-Lef1 (ab22884; Abcam); rabbit polyclonal anti-pan-cytokeratin (sc-15367; Santa Cruz Biotechnology); rabbit monoclonal anti-non-phospho (active)-β-catenin (8814S; Cell Signaling Technology); and rabbit monoclonal anti-Ki67 (ab16667; Abcam).

The secondary antibodies used were as follows: goat anti-rabbit IgG-HRP (SC-2004; Santa Cruz Biotechnology); mouse monoclonal anti-mouse-IgGκ binding protein-HRP (SC-516102; Santa Cruz Biotechnology); donkey polyclonal anti-rabbit IgG (H+L), Alexa Fluor 594 (A-21207; Thermo Fisher Scientific); and donkey polyclonal anti-mouse IgG (H+L), Alexa Fluor 488 (A-21202; Thermo Fisher Scientific).

## Three-dimensional reconstruction and dimension measurement

Serial 5-μm frontal sections of the E60 and E90 miniature pig mandibles and tooth germs were obtained and stained with H&E. Microscopic images of all sections were taken and then aligned using ImageJ software (v1.50i; National Institutes of Health, Bethesda, MD). The outline of the mandible and tooth germ was reconstructed using Amira software (v6.0.1; Thermo Fisher Scientific, Waltham, MA). Widths of the DC and labial and lingual alveolar sockets were measured at the plane of the bottom of the SDL or PC.

### Quantification of mandible deformation after surgery

Fresh miniature pig embryos were obtained, and their mandibles were dissected quickly. Each mandible was fixed on a platform and not moved during surgery and scanning. Then, the first of two micro-CT imaging examinations was performed. After scanning, the gingiva above the canine was cut to release stress. The cut was an incision in the overlying gingival soft tissue, and the overlying gingiva was not removed. Subsequently, the second micro-CT imaging examination was performed. These procedures were finished in half an hour in total. In this way, we obtained micro-CT data of the mandible before and after surgery.

For the "sham surgery" group, the mandible was also dissected and fixed on the platform. After the first micro-CT, the platform with the mandible was taken out of the micro-CT scanner, but the mandible was not cut. The second micro-CT imaging examination followed immediately.

Next, the images of the mandible slice containing the canines before and after surgery were isolated with the software Geomagic Studio (3D Systems, Rock Hill, SC) and then aligned to generate a three-dimensional (3-D) color map with the software Geomagic Control (3D Systems). The cross-section through the tip of the canine was selected to generate a two-dimensional (2-D) image. The solid purple contour and dotted black contour showed the pre-surgery and post-surgery shape, respectively. The distance between the two contours was the colored line segments showing the distance and direction of the movement. The quantification of deformation was determined by measuring the length of the colored line segments (Fig 2D–E). The colored ball in 2-D comparison marked the position of the maximum displacement.

The surgery and micro-CT was done by one researcher, and the data of each sample were labeled with Arabic numerals, which were blinded for the second researcher. The second researcher performed the quantification analysis of the deformation.

### Cup model establishment and calculation of stress value

The quantity of stress was determined based on the deformation and mechanical features of the mandible by establishing a cup model using the finite element analysis software ANSYS (ANSYS, Canonsburg, PA). Mandible deformation quantification, Young's modulus determination, Poisson's ratio evaluation, set of boundary conditions, stress value calculation, and other details regarding the cup model are provided in Appendix Supplementary Methods.

### *In vitro* mandible culture

We dissected the mandible of a miniature pig embryo at E60 followed by cutting at the mesial and distal planes to the DC using microscissors to obtain a mandible slice. Mandible slices were cultured on a Transwell membrane (pore size: 0.4 μm; Corning Inc., Corning, NY) in minimum essential medium α (MEM-α; Thermo Fisher Scientific) supplemented with 15% fetal bovine serum (FBS), 2 mM glutamine, 100 U/ml penicillin, and 100 mg/ml streptomycin. Tissues were incubated at 37°C in an atmosphere containing 5% $CO_2$ for 2 days, and the medium was refreshed every 24 h. Activating Wnt signaling with LiCl, inhibiting Wnt signaling with recombinant protein Dkk1, and *RUNX2* overexpression or knockdown with lentivirus are described in Appendix Supplementary Methods.

### Exerting stress on mandibles *in vitro*

E60 mandible slices were placed into individual wells of BioPress Compression Plates (Flexcell International, Burlington, NC) and cultured in MEM-α supplemented with 15% FBS, 2 mM glutamine, 100 U/ml penicillin, and 100 mg/ml streptomycin. Tissues were incubated at 37°C and 5% $CO_2$ for 48 h, and the medium was refreshed every 12 h.

Static compression of 3 kPa was applied using the FX-5000 Compression System (Flexcell International). Mechanical compression lasted for 48 h and included repeated test cycles. In each cycle, the static 3-kPa compression was applied for 30 min, followed by relaxation (0 kPa) for 5 min. The culture media could penetrate the tissue during the relaxation time. In the control groups, samples were also cultured in BioPress Compression Plates, but no mechanical compression was applied.

### Primary culture of DFCs and inhibition of ERK1/2, integrin β1, and Wnt signaling

Dental follicle tissue was microdissected from the peripheral region of the tooth germ of an E60-miniature pig embryo with a dissecting microscope. The tissue was digested with dispase II (4 mg/ml) and collagenase type I (3 mg/ml) at 37°C for 1 h. The cell slurry was filtered and centrifuged at 300 *g* for 5 min to obtain the cells. Single-cell suspensions were seeded onto culture dishes and cultured with MEM-α supplemented with 15% FBS, 2 mM glutamine, 100 U/ml penicillin, and 100 mg/ml streptomycin. The cells were incubated at 37°C and 5% $CO_2$.

To inhibit ERK1/2, the chemical U0126 (70 nM; HY-12031; MedChem Express, Monmouth Junction, NJ) was added to the medium; the same concentration of DMSO was added to the control group medium. To block integrin β1, a mouse monoclonal antibody against integrin β1 (5 μg/ml; ab30388; Abcam, Cambridge, UK) was added to the medium; the same amount of PBS (10 μl) was added to the control group medium. To inhibit Wnt signaling, IWR-1-endo (5 μM; 10161; Sigma-Aldrich) was added to the medium; the same amount of PBS was added to the control group medium. For all groups, the treatment lasted 2 h before cells were harvested.

### Force loading on DFCs

To study whether the mechanical force regulated the integrin β1-ERK1-RUNX2-Wnt pathway at a cellular level, we applied compressive force to DFCs using the weighted cover glass method (Feng *et al*, 2016). Briefly, a round glass cover was rinsed with PBS and then placed over an 80% confluent cell layer. A metal weight was placed on top of the glass cover to deliver compression of 1.0, 2.0, or 5.0 g/cm$^2$; in the control group, only a glass cover was used with no weight applied. Cells of the experimental and control groups were harvested 0, 1, 2, or 4 h after force application.

### Real-time RT–PCR

E60 pig mandibles that were cultured under 0 kPa or 3 kPa stress for 2 days were harvested, and E60 pig mandibles that were not cultured were obtained. The PC and the surrounding soft tissue were microdissected, and total RNA was extracted (74106; RNeasy

Mini Kit; Qiagen, Hilden, Germany). Reverse transcription was performed using the SuperScript III First-Strand Synthesis System (18080051; Thermo Fisher Scientific). Real-time RT–PCR was run in triplicate (volumes of 20 μl) using the SYBR Green PCR Master Mix (A25742; Applied Biosystems, Foster City, CA) and on a CFX96 Touch Real-Time PCR Detection System (Bio-Rad Laboratories, Hercules, CA), after which melting curve analysis was performed. The relative expression levels of each gene were normalized to *GAPDH* levels and determined by the $2^{-\Delta\Delta C_t}$ method. Forward and reverse primers for *TGFB1* (TGF-β1), *NFKB1* (NF-κB1), *EGR1*, *RUNX2*, *ITGB1* (integrin β1), *ITGB3* (integrin β3), *ITGAV* (integrin αV), and *ITGA2* (integrin α2) are listed in Appendix Table S2.

### Western blotting

After force exertion, lentiviral vector infection, and/or chemical treatment, the DFCs were harvested. Collected cells were lysed in RIPA Lysis and Extraction Buffer (Thermo Fisher Scientific). Cell lysates (10 μg/lane) were loaded and separated by sodium dodecyl sulfate–polyacrylamide gel electrophoresis (SDS–PAGE). The proteins were then transferred to an Immobilon-P polyvinylidene difluoride membrane (Millipore, Burlington, MA) and incubated with primary antibodies at 4°C overnight. After incubation with secondary antibodies at room temperature for 1 h, membranes were treated with Pierce ECL Western Blotting Substrate (Thermo Fisher Scientific), followed by exposure and digital imaging. Western blot analysis of nuclear proteins is detailed in Appendix Supplementary Methods.

Primary antibodies used were as follows: rabbit monoclonal anti-non-phospho (active)-β-catenin (8814S; Cell Signaling Technology, Danvers, MA), rabbit polyclonal anti-Lef1 (ab22884; Abcam), mouse monoclonal anti-RUNX2 (sc-390351; Santa Cruz Biotechnology, Dallas, TX), rabbit monoclonal anti-p-ERK1/2 (MA5-15173; Thermo Fisher Scientific), and rabbit polyclonal anti-β-actin (ab129348; Abcam). Secondary antibodies used were goat anti-rabbit IgG-HRP (SC-2004; Santa Cruz Biotechnology) and goat anti-mouse IgG (H+L), biotin conjugate (SA00004-1; Proteintech, Rosemont, IL).

### Semi-quantification of expression levels

Semi-quantification of ISH, immunostaining, or Western blot expression levels was performed using Image-Pro Plus 6.0 software (Media Cybernetics, Rockville, MD). First, the image was opened with the software, after which the incident level was set as "255" in the standard optical density calibration and "Integrated Optical Density (IOD)" measurement was selected. For fluorescence analysis, the positive areas were selected manually and marked with red regions. Cell count was then determined by manually counting the number of DAPI-positive cells in the image. Finally, after clicking the "count" button, the measurement and statistics data were obtained. For histological analysis, we calculated the "IOD/cell" value; for each experimental group, the calculation was performed using three different histology slides from three different samples. For the quantification of protein levels, we determined the "IOD/cell" value; quantification was performed in triplicate using three different Western blots.

### Statistical analysis

Unpaired *t*-tests were used to compare the dimensions between two groups ($n = 3$ per group), relative expression levels from IF, ISH, or fluorescent ISH data between two groups ($n = 3$ per group), PC epithelium area between two groups ($n = 3$ per group), and relative expression levels of proteins between two groups ($n = 3$ per group).

One-way ANOVA was used to compare RT–qPCR relative expression data between three groups ($n = 3$ per group), relative expression levels from IF or ISH data between three or more groups ($n = 3$ per group), relative expression levels of proteins between four groups ($n = 3$ per group), and the dimensions between five groups ($n = 3$ per group). This was followed by a Newman–Keuls test for post hoc comparisons between two groups.

All data are presented as the means ± SEM. *P* values < 0.05 were considered statistically significant. All statistical analyses were performed using GraphPad Prism (GraphPad Software Inc., La Jolla, CA).

**Expanded View** for this article is available online.

### Acknowledgements

We thank Jianfeng Lei for help with micro-CT imaging, Kun Jiao for animal surgery, and Shiliang Feng for helpful discussion. We thank Sian Xie and Yajun Zhang for their help in testing Young's modulus. This work was funded by Chinese Academy of Medical Sciences Research Unit (No. 2019RU020), Capital Medical University, Beijing Municipality Government grants to S.W. (Beijing Scholar Program—PXM2018_014226_000021, PXM2017_014226_000023, PXM2018_193312_000006_0028S643_FCG PXM2016_014226_000034, PXM2016_014226_000006, PXM2015_014226_000116, PXM2015_014226_000055, PXM2015_014226_000052, PXM2014_014226_000048, PXM2014_014226_000013, PXM2014_014226_000053, Z121100005212004, PXM2013_014226_000055, PXM2013_014226_000021, and TJSHG201310025005), and the National Natural Science Foundation of China (No. 91649124 to S.W., 81400478 to X.W., and 31661143044 to M.L.).

### Author contributions

XW, GL, Yan L, and JZ performed the histological, cellular, and *in vitro* experiments. JH, SL, and ML performed the CUP modeling and mechanical analysis. Yang L, LH, and GD collected the samples. FW, AL, ZF, CZ, and JW analyzed the data and reviewed the manuscript. SW designed and supervised the project. XW, JH, ML, and SW wrote the manuscript.

### Conflict of interest

The authors declare that they have no conflict of interest.

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
