## [Review Process File · The EMBO Journal]

Biomechanical stress regulates mammalian tooth replacement via the integrin β 1-RUNX2-Wnt pathway

Xiaoshan Wu, Jinrong Hu, Guoqing Li, Yan Li, Yang Li, Jing Zhang, Fu Wang, Ang Li, Lei Hu, Zhipeng Fan, Shouqin Lü, Gang Ding, Chunmei Zhang, Jinsong Wang, Mian Long and Songlin Wang

Review timeline:

Submission date:	1st May 2019
Editorial Decision:	26th Jun 2019
Revision received:	19th Sep 2019
Editorial Decision:	10th Oct 2019
Revision received:	18th Oct 2019
Editorial Decision:	11th Nov 2019
Revision received:	18th Nov 2019
Accepted:	21st Nov 2019

Editor: Ieva Gailite

Transaction Report:

1st Editorial Decision

26th Jun 2019

Thank you for submitting your manuscript for consideration by the EMBO Journal. We have now received three referee reports on your manuscript, which are included below for your information.

Based on the overall interest expressed in the referee reports and the revision outline you provided during the pre-decision discussion, I would like to invite you to submit a revised version of the manuscript in which you address the comments of reviewers #2 and #3 along the lines indicated in the pre-decision consultation, but especially focusing on experimentally addressing the points 5, 11 and 12 by reviewer #2. I should add that it is The EMBO Journal policy to allow only a single major round of revision and that it is therefore important to resolve the main concerns at this stage.

REFeree REPORTS:

Referee #1:

This work addresses challenging problems/questions in the fields of tooth and bone formation: The cellular and molecular mechanisms of tooth replacement, and the molecular basis of biomechanical stress-regulated bone formation. Novel technologies were developed for analysing the cellular and molecular mechanisms underlying tooth replacement, and for measuring biomechanical stress in bone slices of pig mandibles.

The results are interesting and novel, and demonstrate for the first time (1) that the development of the replacement tooth is regulated by biomechanical stress, and (2) reveal the molecular mechanisms how mechanical stress first prevents the development of replacement tooth (integrin-ERK-Runx2), and how the release of this stress triggers replacement tooth initiation (Runx2-Wnt).

Biomechanical stress is currently an important topic in the bone research field in particular bone remodelling, but the underlying molecular mechanisms are not well understood. Hence this work will interest a large audience in tooth/bone research.

Referee #2:

There is increasing evidence showing that mechanical signals play an important role in regulating various aspects of cell biology. However, how such signal is generated and utilized to control tissue renewal remains an important open question. This is in part due to a lack of robust tools for measuring tissue forces and for perturbing forces without affecting other types of signaling. Given these limitations, the current manuscript has performed a series of biomechanical and computational experiments to demonstrate that unerupted deciduous Guinea pig canines regulate the development of their permanent counterpart through compressive stress. They then further showed that the compressive force controls gene expression of Runx2 and components of Wnt pathway in the dental mesenchyme, which may in turn signals to initiate the development of the permanent canine epithelium, although this later aspect is not well studied.

Overall, this manuscript has presented some very interesting results and novel findings that shed light on the importance of mechanical signaling during tissue development. Some of the biomechanical experiments conducted are technically difficult and it's certainly applaudable that they have successfully carried out these experiments to draw one of their main conclusions showing that compressive stress plays a role in regulating gene expression in tissues. This particular demonstration should be of general interest to the wide readership of EMBO Journal, as similar mechanisms may well exist in other tissues with repetitive growth. However, whether the relief of compressive force is linked to tooth eruption is not proven. In addition, the second part of the story describing the relay of the signal through Runx2 and Wnt is less developed and major questions remain on whether force influences gene expression of Wnt components and/or Wnt signaling activity, as well as how signal is relayed from the mesenchyme to the epithelium. In addition, as most of the experiments were done in ex plants setting, certain key control experiments were not performed to ensure correct interpretation of the results. Lastly, certain claims in the paper, such as differential growth resulting in compression, were not sufficiently proven and further experiments will have to be performed to confirm those conclusions. All of these will be further discussed below.

In sum, this is an interesting paper that explores a very important question in developmental biology, and it has the potential to be impactful. However, there are a few points in the paper that are insufficiently supported by their data and these have to be further addressed before this paper can be considered for publication.

Major comments:

1. The authors claimed that the SDL is quiescent or stationary after its separation from the deciduous canine. However, they didn't look at cell proliferation (BrdU or equivalent) nor cell death to assess the state of these cells. They did perform PCNA staining and some of the cells in SDL were labelled, suggesting that some SDL cells may still be undergoing proliferation. Looking at these basic cellular processes at different time points should give us a better understanding of the growth state of SDL before, during, and after canine eruption.
2. The authors looked at the expression of Shh, Pitx2, Pax9, and Sox2 in SDL, but did not draw any conclusion from those staining, other than saying Sox2-expressing stem cells may be present at the tip of SDL. Shh expression is absent in epithelium, how does that compare to primary dental lamina? Pax9 expression is also extremely weak. I'm also not entirely sure how looking at these markers will add to the main conclusion of the story, especially given that most of them are hardly expressed in SDL or neighbouring mesenchyme.
3. I am not fully convinced that "differential growth" generates stress inside the mandible and this should be further addressed. Firstly, there appears to be tissue between the growing canine and alveolar bones and the canines are replacing that tissue between E60 and E90. As a result, the net volume within the U shaped alveolar bones, based on images provided, may not be changing that

much over time and poses the question whether this could actually generate additional stress. Second, there's no functional study showing that altering the growth rate of the canine results in changes in compressive stress. Thus, the observation of growth rate difference and compressive stress are at most a correlation and not a causation at this stage.

4. Related to my point #3, I am also not fully convinced that eruption results in relieved compression. The authors performed surgery by cutting the top of the gingiva at E60, presumably to mimic the eruption process, but this is not the same as canine eruption at E90. Also, at E90, the canine is significantly bigger and it's not clear from any experiment presented in the paper that the eruption process reduces the compressive stress that was observed at E60. Would doing cutting experiments at several different time points between E60 and E90 reveal a trend that would suggest gradual decrease of compression? Would comparing mandible deformation between normal and delayed eruption (e.g. with bisphosphonate or other methods) help? Or perhaps looking at cell shape changes might provide an indirect measure of this? It's certainly possible that, while there is compressive stress at E60, that stress remains at E90 and the initiation of permanent canine *in vivo* is regulated by other signals, while changes in its development in *ex plants* can be artificially influenced by forces.

5. One of the key experiments in the paper is to compress samples with surgically cut gingiva to show that compression is able to rescue the phenotype. However, one important control experiment is to perform the same procedure on uncut samples. This will ensure that 1) cutting and culturing didn't artificially induce SDL expansion and 2) compressing cut samples is a true rescue, as opposed to an artificial response to applied compression. Uncut samples are expected to have narrow SDL in both non-compressed and compressed environment.

6. The authors quantified immunostaining by dividing intensity over area. Are there the same number of cells within a given area? This is important as compression may increase cell density, thus increasing signal per area, but not necessarily per cell. It might be more relevant to compare average signal per cell. The authors are also encouraged to quantify this by doing western on isolated mesenchymal cells from non-compressed and compressed tissues.

7. In Fig. 4, why is integrin beta 1 perinuclear? The authors should also use an antibody against active form of beta 1 (e.g. 550531 from BD) to see if there is indeed increased integrin beta 1 activity upon compression, and not just increased expression.

8. In the experiment where the authors compressed dissociated dental follicle cells, they reported that there is an increase of p-ERK and RUNX2 staining within 2 hours. Is this activation reversible? Also, can the activation be sustained for longer term? These are relevant, as in tissues, mesenchymal cells are presumed to be under compression until E90 and it is the reduction of compression that triggers loss of Runx2 expression.

9. It's not entirely clear how RUNX2 overexpression and RNAi was performed. Was the virus added to the entire culture? If so, why is RUNX2 only detected in the mesenchyme and not in the epithelium as well? If tissue specific overexpression or knockdown was performed, it should be clarified. If there's no tissue specific overexpression or knockdown, the expression pattern should be explained. Also in this case, the authors should acknowledge that the effect of RUNX2 overexpression can also come from the epithelium. Similarly, on pg 17, it says that "The expression levels of Wnt signaling... after DFCs were infected with a RUNX2 overexpression lentiviral vector", was it just the DFCs? Or the entire tissue was infected. And was it a virus infection or transfection with a vector? These need to be clarified. Lastly, it was not mentioned whether this experiment was done on samples with a cut or not.

10. The authors were using the presence of beta-catenin transcript as a proxy to Wnt signaling activity (interpreting it as either up- or down regulated) and that is not correct. To assess Wnt activity, the authors either have to look at a bone fide Wnt target gene, such as Axin2, or the localization of beta-catenin protein in nucleus vs. cytoplasm. Therefore, under the section "Wnt modulation between the mesenchyme and epithelium regulates PC initiation", the authors can only conclude that the expression of genes encoding beta-catenin or Runx2 changes between E60 and E90, but not the actual Wnt activity in either epithelium or mesenchyme. Also, beta-catenin FISH is hardly detectable in the epithelium at E90 (Fig. 6J), contrary to the ISH results in Fig. 6E. Similarly,

on page 16, the authors said "these findings indicated that upregulation of Wnt signaling in the enamel organ can start PC initiation" but since the effect of LiCl is over the entire tissue and the upregulation of Wnt signaling is in both the epithelium and the mesenchyme, one can't conclude that it's the activation of Wnt signaling in the enamel organ that initiates the process, as it could be equally plausible that it's due to Wnt activation in other tissue types, and PC initiation is secondary to that.

11. On pg 17, the authors found increased nuclear beta catenin in various conditions using western, however, this is somewhat confounding as the total beta-catenin is also increased (fig. 6). As a result, does compression and Runx2 regulate Wnt activity in addition to beta-catenin expression? It's likely that the increase in nuclear beta-catenin is due to increased total beta-catenin expression. The authors can perturb Wnt signaling pathway to see if beta catenin still accumulates in the nucleus. If it does, then that could be a passive result due to overall increase of beta-catenin. The authors should either distinguish between these scenarios or acknowledge these possibilities in texts.

12. In the first paragraph of the discussion, the authors stated "Downregulation of this pathway in the mesenchyme activated Wnt signaling in the PT epithelium and thereby triggered PT development". However, no evidence was presented to indicate this. The authors didn't show that reduction of Runx2 or Wnt activity in the mesenchyme will lead to activation of Wnt activity in the epithelium. Perhaps they can investigate this by looking at nuclear beta catenin in peel epithelium from samples infected with Runx2 RNAi. In general, this part of the paper was not well developed and it remains unclear how mesenchyme signals to the epithelium to initiate its development.

Minor comments:

1. For non-tooth biologists, they may not know what a dental lamina is nor the basic developmental stages of a tooth. Additional information regarding these may help readers better understand the project.
2. The authors should be careful about using the word "quiescent" to describe cells in permanent canine germs. Quiescence would refer to these cells being non-proliferative and in a G0 state. However, these were not tested in the paper and these cells should not be referred as quiescent.
3. Please make sure that genes are italicized.
4. The authors mentioned that the third deciduous incisor had similar dynamics as canines (pg. 5). Please clarify if this refers to in relations to incisor eruption.
5. On page 19, where Davidson, 2018 was cited, please cite the primary source.
6. Font size in Fig 7E and F are too small to read.

Referee #3:

I have now read the manuscript in detail. The authors intent to show that mechanical pressure arrests the development of the permanent tooth and that this mechanical effect is mediated through the expression of RUNX2. I can not detect any major problems with what is being proposed. This is in part due, however, to the fact that some methods and results are not adequately described. This precludes me from providing a more detailed review of the manuscript. I cannot, then, suggest the publication of this manuscript in its present form.

In general the results are innovative, although there is a lack of citation of related work, relevant and the experimental evidence described is solid. However, some of the results and methods are not described with enough detail.

Although the English has been revised by natives and it is, overall, correct, there are many parts of the manuscript that are not understandable. This is usually because the context is not explained or

because there is no explicit explanation of the purpose or significance of some experiments. In other cases the structure of the sentences is way too baroque.

1. The second sentence of the second paragraph of section "Differential growth rates generate..." is a bit strange. A more direct style would make it clearer, "To measure the deformation of the mandible (we) ...". In addition, the authors need to provide some more detail on what they did, how and why. The way it is written it seems as if the CT-scanning itself directly informs about deformation, but this is not the case. It is also important to describe the time scale over which this deformation was measured. This is done latter on in the methods but some brief description should be present in the main text.

2. At the end of the paragraph, the Poisson ratio is measured for the whole mandible? Or for the developing teeth or for what.

3. Next paragraph, first sentence. Lack of context. I assume the color map is a map of the deformation. This needs to be stated explicitly.

4. Major issue. In the fourth paragraph of section "Differential growth...": There is no reason why the reader should know what a Piuma Chiaro Nanoindenter is supposed to do. More context and more explanations are needed in here (there is some description of that in the supplementary but this latter description is not very well written either). The second sentence of the paragraph should be rewritten too. One could write that "It was found that Young's modules ranged..." to make clear that this is a result of the manuscript not a general knowledge. The last sentence of the paragraph is even more problematic. Which is the rational to use a model to evaluate the possible values of Poisson's ratio. This is explained, although poorly, in the supplementary but it is important to describe in the main text why one is doing that, which are the assumptions and how the results contributes to the rest of the manuscript.

5. Major issue. Next paragraph. This paragraph is very unclear. Towards the end of the paragraph it is explained that, or it seems, that the purpose of the cups model is to estimate the range of stress values surrounding the tooth. This should be explained at the beginning of the paragraph. The paragraph should also explain what is the model for, which are its assumptions and which is the result. This is not the case currently. It is also puzzling to mention the software that is used for the model, ANSYS 15.0, but not the basics of the model, like that it is a finite element analysis model. From this paragraph and the rest of the article it is quite evident that the communication between those making the model and those making the experiments has not been as fluid as necessary.

6. Next section, first paragraph, 9th line. Apparently, in only half of the experiments does the mechanical compression inhibit PC initiation. What happens the other half of the times?

7. Seventh paragraph of section "Biomechanical stress regulates ...". Which is the rational of this paragraph. Its connection to the rest of the section needs to be more explicit.

8. The second sentence of the second paragraph of section "Mechanical force regulates..." is totally unclear. How does the IF inform about the optimal value of the force and what does optimal mean in this context.

9. Last paragraph of section "RUNX2 overexpression inhibit...". Which is the rational of this paragraph. Its connection to the rest of the section needs to be more explicit.

10. First two sentences of section "Lateral inhibition". The authors are a bit off track here. Lateral inhibition is usually applied to reaction-diffusion or notch-delta type of mechanisms not to mechanical inhibitions.

11. Delete first paragraph of section "Central role of RUNX2-wnt..." the claims there are not sustained by the manuscript and are not cited adequately.

12. Some citations and some more context are required at the beginning of section "Similarities between replacement..."

13. The authors may want to cite recent related work by Renvoise et al.
14. Figure 2A. It is pointless to present some 3D representations in a single view. The same objects should be plotted from several angles to allow for a better perception of the 3D.
15. Major issue. Figure 2E is totally unclear. There is too much graphical information and a far too poor legend. This needs to be corrected for the relevance and adequacy of this experiment to be evaluable.
16. Figure EV2A and B. Again this needs to be explained in more detail (in here and in the supplementary).
17. Figure EV3A. These 3D representations need to be shown from several different angles.
18. It is quite difficult to see anything in figure EV4A-N.
19. Figure 2J is also rather unclear. It understand that is not an analytical result but a simulation result and then there should be points representing each simulation result, not just coloured areas.
20. The authors should provide a quantification of the variation between the two researchers in the analysis of the deformation induced by surgery.
21. Major issue. The supplementary description of the methods lacks detail. This specially the case for the modelling part. In addition, it is not well structured.
22. The second paragraph of the section "Determination of Poisson's ratio in modeling" needs to be explained in more detail. Quite relevant information seems to be missing.
23. Major issue. The authors should discuss in more detail previous related work. That mechanical pressure from the bone arrests permanent tooth development was already known and then needs to be discussed and cited.

Minor issues:

1. Delete "model" from "hair follicle model" in the 5th line of the introduction.
2. Delete "extrinsic" from the 9th line of the article. It is unclear what extrinsic means in this context and it is certainly not necessary to say it.
3. Second paragraph, the second sentence seems to imply that larger animals have more complex tooth shapes but there is no evidence for that. In addition, this statement adds nothing to the manuscript.
4. In the next sentence, it is unclear what "(resting space)" refers to, as usual some a more explicit context would be necessary.
5. The first sentence of the third paragraph is unnecessarily complex, it can be simplified for clarity.
6. Fifth paragraph of section "Biomechanical stress regulates ...". I do not know to what "transition tendency" refers to. In the last sentence of this paragraph, one should delete the last "signaling"
7. Section "RUNX2 overexpression...". Delete "Meanwhile".

Referee reports:

Referee #1:

This work addresses challenging problems/questions in the fields of tooth and bone formation: The cellular and molecular mechanisms of tooth replacement, and the molecular basis of biomechanical stress-regulated bone formation. Novel technologies were developed for analysing the cellular and molecular mechanisms underlying tooth replacement, and for measuring biomechanical stress in bone slices of pig mandibles. The results are interesting and novel, and demonstrate for the first time (1) that the development of the replacement tooth is regulated by biomechanical stress, and (2) reveal the molecular mechanisms how mechanical stress first prevents the development of replacement tooth (integrin-ERK-Runx2), and how the release of this stress triggers replacement tooth initiation (Runx2-Wnt). Biomechanical stress is currently an important topic in the bone research field in particular bone remodeling, but the underlying molecular mechanisms are not well understood. Hence this work will interest a large audience in tooth/bone research.

We appreciate reviewer 1's positive feedback on our work.

Referee #2:

There is increasing evidence showing that mechanical signals play an important role in regulating various aspects of cell biology. However, how such signal is generated and utilized to control tissue renewal remains an important open question. This is in part due to a lack of robust tools for measuring tissue forces and for perturbing forces without affecting other types of signaling. Given these limitations, the current manuscript has performed a series of biomechanical and computational experiments to demonstrate that unerupted deciduous Guinea pig canines regulate the development of their permanent counterpart through compressive stress. They then further showed that the compressive force controls gene expression of Runx2 and components of Wnt pathway in the dental mesenchyme, which may in turn signals to initiate the development of the permanent canine epithelium, although this later aspect is not well studied.

Overall, this manuscript has presented some very interesting results and novel findings that shed light on the importance of mechanical signaling during tissue development. Some of the biomechanical experiments conducted are technically difficult and it's certainly applaudable that they have successfully carried out these experiments to draw one of their main conclusions showing that compressive stress plays a role in regulating gene expression in tissues. This particular demonstration should be of general interest to the wide readership of EMBO Journal, as similar mechanisms may well exist in other tissues with repetitive growth. However, whether the relief of compressive force is linked to tooth eruption is not proven. In addition, the second part of the story describing the relay of the signal through Runx2 and Wnt is less developed and major questions remain on whether force influences gene expression of Wnt components and/or Wnt signaling activity, as well as how signal is relayed from the mesenchyme to the epithelium. In addition, as most of the experiments were done in ex plants setting, certain key control experiments were not performed to ensure correct interpretation of the results. Lastly, certain claims in the paper, such as differential growth resulting in compression, were not sufficiently proven and further experiments will have to be performed to confirm those conclusions. All of these will be further discussed below.

In sum, this is an interesting paper that explores a very important question in developmental biology, and it has the potential to be impactful. However, there are a few points in the paper that are insufficiently supported by their data and these have to be further addressed before this paper can be considered for publication.

We appreciate reviewer 2's positive feedback and constructive criticism. We have performed additional experiments and revised our manuscript to address all of these issues. The major changes are labeled in blue font in the revised manuscript.

Major comments:

1. The authors claimed that the SDL is quiescent or stationary after its separation from the deciduous canine. However, they didn't look at cell proliferation (BrdU or equivalent) nor cell death to assess the state of these cells. They did perform PCNA staining and some of the cells in SDL were labelled, suggesting that some SDL cells may still be undergoing proliferation. Looking at these basic cellular processes at different time points should give us a better understanding of the growth state of SDL before, during, and after canine eruption.

Thank you for this helpful suggestion. We performed additional experiments to compare cell proliferation and apoptotic state of the SDL before, during, and after canine eruption using Ki67 staining and a TUNEL assay. As the Brdu assay is not suitable for large mammals including miniature pigs, we used Ki67 staining instead of the Brdu assay to study cell proliferation. The data further indicated that the SDL is resting after its separation from the deciduous canine in Fig 1I-K of the revised manuscript.

2. The authors looked at the expression of Shh, Pitx2, Pax9, and Sox2 in SDL, but did not draw any conclusion from those staining, other than saying Sox2-expressing stem cells may be present at the tip of SDL. Shh expression is absent in epithelium, how does that compare to primary dental lamina? Pax9 expression is also extremely weak. I'm also not entirely sure how looking at these markers will add to the main conclusion of the story, especially given that most of them are hardly expressed in SDL or neighbouring mesenchyme.

Thank you for the helpful comments. We performed *in situ* hybridization of Shh on primary dental lamina (E40) and found that Shh is absent (data is shown in Appendix Fig S2). Indeed, we did not draw any conclusions based on the expression of these markers since the staining results were not entirely relevant to the main conclusion of the study. As a result, we kept only the Sox2 expression in Fig 1H and moved most of the staining results into the appendix files (Fig EV1C-E) as a reference pattern for these markers.

3. I am not fully convinced that "differential growth" generates stress inside the mandible and this should be further addressed. Firstly, there appears to be tissue between the growing canine and alveolar bones and the canines are replacing that tissue between E60 and E90. As a result, the net volume within the U shaped alveolar bones, based on images provided, may not be changing that much over time and poses the question whether this could actually generate additional stress. Second, there's no functional study showing that altering the growth rate of the canine results in changes in compressive stress. Thus, the observation of growth rate difference and compressive stress are at most a correlation and not a causation at this stage.

Thank you for the helpful comments. The space between the canine and alveolar bone is occupied by tissue, mainly loose connective tissue, or fluid inside the dental sac. These tissues are mainly thought to act as transfer media for biomechanical force. As the net volume within the U shape does not change substantially over time, the rapid growth of the deciduous canine may generate stress within the sealed mandible. The eruption of deciduous canine breaks the seal surrounding the mandible and subsequently the stress is reduced. These results support that differential growth might generate stress inside the mandible.

There was no functional study showing growth rate results with changes in compressive stress. We agree that the observation of growth rate differences and changes in compressive stress are at most a correlation and do not signify causation. We have revised the manuscript accordingly. Specifically, we revised the title of the section "Differential growth rates generate biomechanical stress inside the mandible" to "Biomechanical stress is generated inside the mandible," and added our assumption that the differential growth rates might generate biomechanical stress inside the mandible in the revised manuscript.

4. Related to my point #3, I am also not fully convinced that eruption results in relieved compression. The authors performed surgery by cutting the top of the gingiva at E60,

presumably to mimic the eruption process, but this is not the same as canine eruption at E90. Also, at E90, the canine is significantly bigger and it's not clear from any experiment presented in the paper that the eruption process reduces the compressive stress that was observed at E60. Would doing cutting experiments at several different time points between E60 and E90 reveal a trend that would suggest gradual decrease of compression? Would comparing mandible deformation between normal and delayed eruption (e.g. with bisphosphonate or other methods) help? Or perhaps looking at cell shape changes might provide an indirect measure of this? It's certainly possible that, while there is compressive stress at E60, that stress remains at E90 and the initiation of permanent canine *in vivo* is regulated by other signals, while changes in its development in ex plants can be artificially influenced by forces.

Thank you for the helpful comments and suggestions. We agree with the suggested experiments proposed by reviewer. We have performed "cut experiments" on the mandible of E65 and E75. The stress ranges of these two stages were evaluated with corresponding mean values of Young's modulus (E65=2.1 MPa, E75=3.7 MPa), and that of E60 was also evaluated for comparison using the mean value of Young's modulus (E60=0.33 MPa).

The results showed that the stress was 8.2-15.1 kPa for E60, 6.8-8.6 kPa for E65, and 10.8-12.7 kPa for E75. Here non-monotonic tendency of gradual stress releasing or increasing could be seen with $E60 > E65 < E75$. In physical point, it is reasonable because the stress is determined upon both Young's modulus and deformation based on the similar relationship of Hooke's law. In combination of the increase of mean Young's modulus from 0.33 MPa of E60, 2.1 MPa of E65 to 3.7 MPa of E75, and the decrease of deformation from of 36.48~79.74 μm E60, 6.86~24.29 μm of E65 to 3.45~14.08 μm of E75, the resulted stress could present non-monotonic tendency which depended on the change extents of both Young's modulus and deformation. In addition, we could not exclude those experimental errors in measuring Young's modulus or deformation, especially for the results of E60 with small Young's modulus and large deformation that covered the elaborative change tendency of stress. The complementary data was shown in Appendix Fig S4 and S5.

When the eruption process starts at E85, the dental sac becomes broken (Fig 1D). This histological finding supports that the pressure from dental sac of DC is decreased dramatically upon eruption. In conclusion, we find that the stress is maintained prior to eruption.

In addition, we agree that it is possible that the initiation of permanent canine *in vivo* can be regulated by other signals, while changes in its development in ex plants can be artificially influenced by forces. As a result, we have revised the discussion part of the revised manuscript accordingly.

The assessment of stress in the mandible of E65 and E75. E60 was also shown for reference.

5. One of the key experiments in the paper is to compress samples with surgically cut gingiva to show that compression is able to rescue the phenotype. However, one important control experiment is to perform the same procedure on uncut samples. This will ensure that 1) cutting and culturing didn't artificially induce SDL expansion and 2) compressing cut samples is a true rescue, as opposed to an artificial response to applied compression. Uncut samples are expected to have narrow SDL in both non-compressed and compressed environment.

Thank you for the comments and suggestions. We regret that we did not describe the experimental process clearly in the previous manuscript. To determine the possible stress inside the mandible, we made a cut on the top of the gingiva of the canine in the whole mandible to release the stress and then measured the deformation using computational three-dimensional reconstruction. However, in the experiments of mandible culture *in vitro*, we did not make a cut on the top of gingiva of the canine mandible segment in both the compressed and non-compressed control samples in the previous version of the manuscript. We cut the canine mandible segment by cutting the medial and distal gingival points of the DC and harvested the mandible segment, where the sealed mandible environment was disturbed. We found that the development of permanent canine was initiated in the control samples and inhibited in the compression samples. We regret that the previous schema (Fig 3A) was misleading, as there was no gingiva on the top of the mandible slice. We have corrected it in the Fig 3A of the revised manuscript.

Yes, we agree that cut or uncut gingiva may influence the results. Therefore, we performed compression experiments on both the cut and uncut mandible segment samples. We found that the SDL was inhibited in both the cut and uncut gingiva samples. The data is shown in Appendix Fig S7.

Ideally, it is better to culture the entire mandible sample to check the SDL in both non-compressed and compressed environments *in vitro*. The entire mandible is too large to be cultured *in vitro*. Thus, it is impossible to observe the SDL on the uncut whole mandible samples cultured *in vitro*.

The SDL was inhibited both in gingiva cut and uncut samples

6. The authors quantified immunostaining by dividing intensity over area. Are there the same number of cells within a given area? This is important as compression may increase cell density, thus increasing signal per area, but not necessarily per cell. It might be more relevant to compare average signal per cell. The authors are also encouraged to quantify this by doing western on isolated mesenchymal cells from non-compressed and compressed tissues.

Thank you for the suggestions. We agree that it is better to compare average signal per cell to quantify the immunostaining data. We performed the quantification by comparing average signal per cell, both in western blotting and immunostaining figures; the data are shown in the revised manuscript.

7. In Fig. 4, why is integrin beta 1 perinuclear? The authors should also use an antibody against active form of beta 1 (e.g. 550531 from BD) to see if there is indeed increased integrin beta 1 activity upon compression, and not just increased expression.

Thank you for the helpful suggestion. Integrin beta 1 is actually expressed at the cell membrane. The perinuclear pattern may be an artifact.

We purchased the antibody against the active form of beta 1 (550531 from BD) and found that it did not react with the species of pig used in our model. We have as yet not found any other antibody against the active form of integrin beta1 which reacts in our pig model.

8. In the experiment where the authors compressed dissociated dental follicle cells, they reported that there is an increase of p-ERK and RUNX2 staining within 2 hours. Is this activation reversible? Also, can the activation be sustained for longer term? These are relevant, as in tissues, mesenchymal cells are presumed to be under compression until E90 and it is the reduction of compression that triggers loss of Runx2 expression.

Thank you for your comments. To determine whether the activation could be reversed or sustained after the reduction of compression, we performed additional experiments: first we compressed the cells for 2 hours then removed the force (group of RFs) and cultured for another 1, 2, or 4 hours before harvesting cells. With an immunofluorescence assay, we found that the expression level of RUNX2 could be sustained for 2 hours, while the expression level of p-ERK1/2 could be sustained for only 1 hour after removing the force. The activation was downregulated for both at the 4th hour after removing the force. These results have been added in Fig 4U-V of the revised manuscript.

9. It's not entirely clear how RUNX2 overexpression and RNAi was performed. Was the virus added to the entire culture? If so, why is RUNX2 only detected in the mesenchyme and not in the epithelium as well? If tissue specific overexpression or knockdown was performed, it should be clarified. If there's no tissue specific overexpression or knockdown, the expression pattern should be explained. Also in this case, the authors should acknowledge that the effect of RUNX2 overexpression can also come from the epithelium. Similarly, on pg 17, it says that "The expression levels of Wnt signaling... after DFCs were infected with a RUNX2 overexpression lentiviral vector", was it just the DFCs? Or the entire tissue was infected. And was it a virus infection or transfection with a vector? These need to be clarified. Lastly, it was not mentioned whether this experiment was done on samples with a cut or not.

Thank you for your comments. The virus was added to the entire media. The specific expression pattern occurred because the mesenchyme cells were easily affected due to the loose cell connections. We acknowledge that the effect of RUNX2 overexpression may also come from the epithelium and have revised the text accordingly.

In response to the other questions, it only involved the cultured DFCs, not the cells separated from tissue. It was a viral infection. This experiment was done on samples after dissection, but without a gingiva cut. We have clarified all relevant information in the text of the revised manuscript.

10. The authors were using the presence of beta-catenin transcript as a proxy to Wnt signaling activity (interpreting it as either up- or down regulated) and that is not correct. To assess Wnt activity, the authors either have to look at a bone fide Wnt target gene, such as Axin2, or the localization of beta-catenin protein in nucleus vs. cytoplasm. Therefore, under the section "Wnt modulation between the mesenchyme and epithelium regulates PC initiation", the authors can only conclude that the expression of genes encoding beta-catenin or Runx2 changes between E60 and E90, but not the actual Wnt activity in either epithelium or mesenchyme. Also, beta-catenin FISH is hardly detectable in the epithelium

at E90 (Fig. 6J), contrary to the ISH results in Fig. 6E. Similarly, on page 16, the authors said "these findings indicated that upregulation of Wnt signaling in the enamel organ can start PC initiation" but since the effect of LiCl is over the entire tissue and the upregulation of Wnt signaling is in both the epithelium and the mesenchyme, one can't conclude that it's the activation of Wnt signaling in the enamel organ that initiates the process, as it could be equally plausible that it's due to Wnt activation in other tissue types, and PC initiation is secondary to that.

Thank you for the comments and suggestions. We performed additional experiments analyzing the localization of the beta-catenin protein in the nucleus vs. the cytoplasm and performed quantitative assessments via RNAscope (RNA ISH at the single molecule level within the cell that identifies subcellular location of signal in the nucleus and cytoplasm) (Arneson et al, 2018; Schulz et al, 2018). The quantification results showed that during the PC initiation from E60 to E90, the nuclear beta-catenin signal in the mesenchyme decreased approximately 10-fold, while the nuclear beta-catenin signal in the epithelium increased approximately 3-fold. In the mandible explant *in vitro*, a similar tendency in both the epithelium and mesenchyme was observed after 12 hours in culture. As a result, these results support that finding that Wnt signals are relayed from the mesenchyme to the epithelium.

We have revised the manuscript and the results showing that beta-catenin was expressed strongly at the epithelium of the PC (new Fig 6J). We agree with the reviewer that we cannot conclude from the LiCl experiment that the activation of Wnt signaling in the enamel organ is what starts PC initiation. We have altered this part accordingly in the revised manuscript.

References

1. Arneson D, Zhang G, Ying Z, Zhuang Y, Byun HR, Ahn IS, Gomez-Pinilla F, Yang X (2018) Single cell molecular alterations reveal target cells and pathways of concussive brain injury. *Nat Commun* 9: 3894
2. Schulz D, Zanutelli VRT, Fischer JR, Schapiro D, Engler S, Lun XK, Jackson HW, Bodenmiller B (2018) Simultaneous Multiplexed Imaging of mRNA and Proteins with Subcellular Resolution in Breast Cancer Tissue Samples by Mass Cytometry. *Cell Syst* 6: 531

The quantification of beta-catenin in the nucleus vs. cytoplasm with assay of RNAscope

11. On pg 17, the authors found increased nuclear beta catenin in various conditions using western, however, this is somewhat confounding as the total beta-catenin is also increased (fig. 6). As a result, does compression and Runx2 regulate Wnt activity in addition to beta-catenin expression? It's likely that the increase in nuclear beta-catenin is due to increased total beta-catenin expression. The authors can perturb Wnt signaling pathway to see if beta catenin still accumulates in the nucleus. If it does, then that could be a passive result due

to overall increase of beta-catenin. The authors should either distinguish between these scenarios or acknowledge these possibilities in texts.

Thank you for the comments and suggestions. To study whether the increase in nuclear beta-catenin is due to Wnt activation or increased total beta-catenin, we performed additional tests, adding the Wnt inhibitor IWR-1-endo to the culture system. The results showed that nuclear beta-catenin was upregulated in the compression and RUNX2 over-expression groups and downregulated after the inhibitor was added. These findings indicate that the increase in nuclear beta-catenin is due to Wnt activation. The data is shown in Fig 7J of the revised manuscript.

12. In the first paragraph of the discussion, the authors stated "Downregulation of this pathway in the mesenchyme activated Wnt signaling in the PT epithelium and thereby triggered PT development". However, no evidence was presented to indicate this. The authors didn't show that reduction of Runx2 or Wnt activity in the mesenchyme will lead to activation of Wnt activity in the epithelium. Perhaps they can investigate this by looking at nuclear beta catenin in peel epithelium from samples infected with Runx2 RNAi. In general, this part of the paper was not well developed and it remains unclear how mesenchyme signals to the epithelium to initiate its development.

Thank you for the comments and suggestions. The SDL is a long and thin epithelial band embedded inside the mandible. It is impossible to separate it from the surrounding mesenchyme under a microscope. As shown in Appendix Fig S6A-G, the operator can only dissect out the PC (SDL) together with the surrounding mesenchyme. As a result, we cannot separate the SDL and quantify the expression level by extracting protein. Therefore, quantification of nuclear beta-catenin with RNAscope is practical.

To study how the Wnt signal is relayed from the mesenchyme to the epithelium, we studied the RNA expression level of beta-catenin in the nucleus vs. the cytoplasm from samples infected with RUNX2 shRNA lentivirus using RNAscope. The results showed that the downregulation of RUNX2 lead to the activation of Wnt signaling in the epithelium. The signal is stronger than that of the scramsh group (Fig 6L-M of the revised manuscript)

We agree that we do not have strong evidence that the reduction of Runx2 or Wnt activity in the mesenchyme leads to the activation of Wnt activity in the epithelium. This needs to be further investigated. We have revised the statement accordingly and discussed the possible mechanism in the discussion section of the revised manuscript.

Minor comments:

1. For non-tooth biologists, they may not know what a dental lamina is nor the basic developmental stages of a tooth. Additional information regarding these may help readers better understand the project.

Thank you. We have added the additional information regarding the basic developmental stages of a tooth in the revised manuscript.

2. The authors should be careful about using the word "quiescent" to describe cells in permanent canine germs. Quiescence would refer to these cells being non-proliferative and in a G0 state. However, these were not tested in the paper and these cells should not be referred as quiescent.

Thank you. We have modified this word into "resting" in the revised manuscript.

3. Please make sure that genes are italicized.

Thank you. We have corrected the genes to be italicized.

4. The authors mentioned that the third deciduous incisor had similar dynamics as canines (pg. 5). Please clarify if this refers to in relations to incisor eruption.

We have clarified the relations to incisor eruption in the revised manuscript.

5. On page 19, where Davidson, 2018 was cited, please cite the primary source.

Thank you. We cited the primary source in the revised manuscript.

6. Font size in Fig 7E and F are too small to read.

We have adjusted the font size in these figures.

Referee #3

I have now read the manuscript in detail. The authors intent to show that mechanical pressure arrests the development of the permanent tooth and that this mechanical effect is mediated through the expression of RUNX2. I can not detect any major problems with what is being proposed. This is in part due, however, to the fact that some methods and results are not adequately described. This precludes me from providing a more detailed review of the manuscript. I cannot, then, suggest the publication of this manuscript in its present form.

In general the results are innovative, although there is a lack of citation of related work, relevant and the experimental evidence described is solid. However, some of the results and methods are not described with enough detail.

Although the English has been revised by natives and it is, overall, correct, there are many parts of the manuscript that are not understandable. This is usually because the context is not explained or because there is no explicit explanation of the purpose or significance of some experiments. In other cases the structure of the sentences is way too baroque.

Thank you for the comments and suggestions. We have revised the manuscript carefully, especially the explanation of the significance of some experiments. We have also added additional citations of related work.

1. The second sentence of the second paragraph of section "Differential growth rates generate..." is a bit strange. A more direct style would make it clearer, "To measure the deformation of the mandible (we) ...". In addition, the authors need to provide some more detail on what they did, how and why. The way it is written it seems as if the CT-scanning itself directly informs about deformation, but this is not the case. It is also important to describe the time scale over which this deformation was measured. This is done latter on in the methods but some brief description should be present in the main text.

Thank you for the comments and suggestions. We have revised the title of the section to "Biomechanical stress is generated inside the mandible".

In the beginning of the second paragraph of the revised manuscript, we have described the time frame during which the deformation and mechanical stress were measured. We have provided more detailed information on the purpose and methods in the revised manuscript.

2. At the end of the paragraph, the Poisson ratio is measured for the whole mandible? Or for the developing teeth or for what.

Sorry for the confusion raised. Actually, no experiments were used to obtain Poisson's ratio; we instead used computational simulations to evaluate the impact of a series of Poisson ratios (0.15, 0.35, and 0.48) on the simulation results (Fig EV2F'). As Poisson's ratio only slightly alters the deformation of the walls of the mandible, a value of 0.35 was finally chosen for all simulations, which is also within a common range for mechanical simulations for cartilage. As Poisson's ratio only slightly alters the deformation of the walls of mandible, a value of 0.35 was finally chosen for all simulations. We have revised the

statement in the fifth paragraph and provide more detailed information in the methods of the revised manuscript.

3. Next paragraph, first sentence. Lack of context. I assume the color map is a map of the deformation. This needs to be stated explicitly.

Yes, the color map is a map of the deformation of the mandible, which we have stated explicitly in the third paragraph of the revised manuscript.

4. Major issue. In the fourth paragraph of section "Differential growth...": There is no reason why the reader should know what a Piuma Chiaro Nanoindenter is supposed to do. More context and more explanations are needed in here (there is some description of that in the supplementary but this latter description is not very well written either). The second sentence of the paragraph should be rewritten too. One could write that "It was found that Young's modulus ranged..." to make clear that this is a result of the manuscript not a general knowledge. The last sentence of the paragraph is even more problematic. Which is the rationale to use a model to evaluate the possible values of Poisson's ratio. This is explained, although poorly, in the supplementary but it is important to describe in the main text why one is doing that, which are the assumptions and how the results contribute to the rest of the manuscript.

Thank you for the helpful comments and suggestions. Briefly, 1) we have moved the Nanoindenter part to the fifth paragraph, and added more explanations for why we need to use this device. We have refined the descriptions in the supplementary materials; 2) we have added "It was found that" at the beginning of that sentence to show how this related to the manuscript; 3) we treated the mandible as a homogeneous and elastic continuum in our modeling. As experimental measurements of the Poisson's ratio in such diminutive samples are complex, computational simulations were used to evaluate the impact of different settings of Poisson's ratio (0.15, 0.35 and 0.48) on the simulation results (Fig EV2F'). As Poisson's ratio only slightly alters the deformation of the walls of the mandible, a value of 0.35 was finally chosen for all simulations, since it's a medium value of cartilage Poisson's ratio (0.15-0.45) and also within a common range for mechanical simulations for cartilage. We have added more descriptions on why and how to evaluate the possible value of Poisson's ratio in the fifth paragraph. We have provided more detailed information on the methods and the contributions of Young's modulus and Poisson's ratio in the revised manuscript.

5. Major issue. Next paragraph. This paragraph is very unclear. Towards the end of the paragraph it is explained that, or it seems, that the purpose of the cups model is to estimate the range of stress values surrounding the tooth. This should be explained at the beginning of the paragraph. The paragraph should also explain what is the model for, which are its assumptions and which is the result. This is not the case currently. It is also puzzling to mention the software that is used for the model, ANSYS 15.0, but not the basics of the model, like that it is a finite element analysis model. From this paragraph and the rest of the article it is quite evident that the communication between those making the model and those making the experiments has not been as fluid as necessary.

Thank you for the helpful comments and suggestions. We have described the purpose of the cup model in the second paragraph, then the details in the fourth and fifth paragraph in the revised manuscript. We have added necessary information about the software ANSYS 15.0, including "finite element analysis (FEA) software" and the assumption that "the mandible was set as a homogeneous and elastic continuum". We have also discussed our modeling description more carefully.

6. Next section, first paragraph, 9th line. Apparently, in only half of the experiments does the mechanical compression inhibit PC initiation. What happens the other half of the times?

Thank you for your comments. The other half of mandible samples failed to get any conclusion in the following conditions. 1) under the compression, the PC was pushed to

outside of frame of mandible slice, and thus the PC was not actually compressed. 2) under the compression, the apoptosis of PC was induced probably due to the poor infiltration of culture media.

7. Seventh paragraph of section "Biomechanical stress regulates ...". Which is the rationale of this paragraph. Its connection to the rest of the section needs to be more explicit.

Thank you. We have modified the first sentence of the seventh paragraph and made the connection to be more clear and explicit in the revised manuscript.

8. The second sentence of the second paragraph of section "Mechanical force regulates..." is totally unclear. How does the IF inform about the optimal value of the force and what does optimal mean in this context.

In the second paragraph of this section of the revised manuscript, we have explained how to get the optimum value with IF assay and explained what the optimal means.

9. Last paragraph of section "RUNX2 overexpression inhibits...". Which is the rationale of this paragraph. Its connection to the rest of the section needs to be more explicit.

Thank you. We have modified the first sentence of this paragraph to make the connection more explicit in the revised manuscript.

10. First two sentences of section "Lateral inhibition". The authors are a bit off track here. Lateral inhibition is usually applied to reaction-diffusion or notch-delta type of mechanisms not to mechanical inhibitions.

Thank you for your comment. "Lateral inhibition" should not be discussed here. We have deleted the first two sentences and combined the rest of this section into the above section in the revised manuscript.

11. Delete first paragraph of section "Central role of RUNX2-wnt..." the claims there are not sustained by the manuscript and are not cited adequately.

Thank you for your comment. We agree and have deleted that paragraph in the revised manuscript and we have deleted the word "niche" in the section title.

12. Some citations and some more context are required at the beginning of section "Similarities between replacement..."

Thank you for your suggestion. We have included more citations and more context in this part.

13. The authors may want to cite recent related work by Renvoise et al.

Thank you for your comment. We have reviewed the recent work by Renvoise et al and cited it in the third discussion paragraph of the revised manuscript.

14. Figure 2A. It is pointless to present some 3D representations in a single view. The same objects should be plotted from several angles to allow for a better perception of the 3D.

Thanks for suggestion. We have added other views from several angles. Please see the Figure below and Appendix Fig S3.

Views of the three-dimensional reconstruction from different angles

15. Major issue. Figure 2E is totally unclear. There is too much graphical information and a far too poor legend. This needs to be corrected for the relevance and adequacy of this experiment to be evaluable.

Thank you for your suggestion. We have revised Fig 2E and provided more information in the legend carefully.

16. Figure EV2A and B. Again this needs to be explained in more detail (in here and in the supplementary).

Thank you for the suggestion. We have explained Figure EV2A and B in more detail in the revised manuscript.

17. Figure EV3A. These 3D representations need to be shown from several different angles.

Thank you for the suggestion. We have added other views from several angles. The data was shown in new Fig EV3A.

18. It is quite difficult to see anything in figure EV4A-N.

Thank you for your comment. Actually, the conclusion from this result is similar to that of Fig 4A-L. This result shows the staining of ISH, while Fig 4A-L shows the staining of IF. Because we cannot gain more conclusions from this part, we deleted Fig EV4A-L in the revised manuscript.

19. Figure 2J is also rather unclear. It understand that is not an analytical result but a simulation result and then there should be points representing each simulation result, not just coloured areas.

Thank you for your comments. We have presented a sample of simulation in the section of "boundary conditions in modeling and interpretation of simulation results" in the Appendix materials and methods. Actually, inclined dashed and solid lines in Figure 2J was plotted based on series of simulation results (many data points). As those points aligned extremely like straight lines, which was caused by small and linear deformations of the system, the

points were omitted for clarity and concise. Further annotation was added to Figure 2J caption.

20. The authors should provide a quantification of the variation between the two researchers in the analysis of the deformation induced by surgery.

Thank you for your suggestion. We are sorry that we did not clarify clearly. In the analysis of the deformation, the two researchers had different jobs. The surgery and micro-CT was done by first researcher, and the data of each sample were labeled with Arabic numerals, which were blinded for the second researcher. The second researcher performed the quantification analysis of the deformation. We have modified the corresponding text in the Appendix materials and methods in the revised manuscript.

21. Major issue. The supplementary description of the methods lacks detail. This specially the case for the modelling part. In addition, it is not well structured.

Thank you for your comment. We have revised the supplementary methods and include more detail.

22. The second paragraph of the section "Determination of Poisson's ratio in modeling" needs to be explained in more detail. Quite relevant information seems to be missing.

Thank you for your comment. Actually, as aforementioned, we did not have experiments for the mandible sample, but the impact of different settings of Poisson's ratio (0.15, 0.35 and 0.48) on the simulation results were evaluated upon computational simulations (Fig EV2F'). As Poisson's ratio only slightly alters the deformation of the walls of mandible, a value of 0.35 was finally chosen for all simulations, and this value was also within a common range for mechanical simulations for cartilage. We have added the necessary information for the section "Determination of Poisson's ratio in modeling" and changed the title into "Evaluation of varied Poisson's ratio in modeling" in the revised manuscript.

23. Major issue. The authors should discuss in more detail previous related work. That mechanical pressure from the bone arrests permanent tooth development was already known and then needs to be discussed and cited.

Thank you for the suggestion. We have cited two more related works and discuss the previous findings in more detail in the third paragraph of discussion.

Minor issues:

1. Delete "model" from "hair follicle model" in the 5th line of the introduction.

We have deleted this word.

2. Delete "extrinsic" from the 9th line of the article. It is unclear what extrinsic means in this context and it is certainly not necessary to say it.

We have deleted this word.

3. Second paragraph, the second sentence seems to imply that larger animals have more complex tooth shapes but there is no evidence for that. In addition, this statement adds nothing to the manuscript.

We have deleted the words of describing "complex tooth shape" in the revised manuscript.

4. In the next sentence, it is unclear what "(resting space)" refers to, as usual some a more explicit context would be necessary.

We have deleted the "(resting space)" in the sentence, and added more explanation in this paragraph.

5. The first sentence of the third paragraph is unnecessarily complex, it can be simplified for clarity.

We have revised this sentence.

6. Fifth paragraph of section "Biomechanical stress regulates ...". I do not know to what "transition tendency" refers to. In the last sentence of this paragraph, one should delete the last "signaling"

We have revised the context to be more explicit.

7. Section "RUNX2 overexpression...". Delete "Meanwhile".

We have deleted this word.

2nd Editorial Decision

10th Oct 2019

Thank you for submitting a revised version of your manuscript. The study has now been seen by two of the original referees. While both reviewers appreciate the added information, they also find that additional minor experiments and textual explanations are needed before they can recommend publication of the manuscript. Therefore, I would like to invite you to submit a revised manuscript, addressing the remaining referee comments, especially regarding providing additional data to demonstrate increased nuclear localisation of beta-catenin in the mandible (referee #2, point 2). Please also address the following editorial issues.

REFeree REPORTS:

Referee #2:

In this revised manuscript, the authors have addressed all my questions and most of my concerns have been answered satisfactorily. There are a couple more points that need some clarification (see below) and should be addressed. I maintain my view that the story is both interesting and important and upon the revision, the scientific rigor has been met. The manuscript enhances our understanding of how tissue mechanics contributes to tissue development and the findings also have a more general application to other epithelial organ systems. Therefore, if the couple points below can be fully addressed, I would recommend this paper to be published at EMBO.

Major comments:

1. This is regarding my comment 8 in the previous review, where I asked the authors whether compressed dissociated dental follicle cells can "sustain" the expression p-ERK and RUNX2 DURING continued compression. The authors have interpreted that as how long the cell can sustain these expression AFTER compression. The answer to this is already provided in the revised manuscript as they've shown that after 4 hours of compression, p-ERK and RUNX2 stain in are both starting to decrease. I thought this is somewhat interesting, given that the expression of these proteins are maintained in vivo, as tissues are under constant compression. Can the authors point this out and discuss why expression cannot be maintained by continuous compression in culture? Perhaps the maintenance requires additional signals?

2. So this could be my ignorance, but I am only aware that it is the level of the nuclear beta-catenin protein that is used as an indicator of active Wnt signaling, and not the beta-catenin RNA level in the nuclei. If there is a paper that shows differential distribution of beta-catenin RNA in nuclei vs cytoplasm in response to Wnt signaling, please provide the citation for that. Otherwise, the authors really have to show beta-catenin antibody staining, as opposed to in situ or RNA scope, in order to claim activated Wnt signal. Otherwise, the RNA level merely suggests that those cells are capable of Wnt signaling as beta-catenin expression is increased, and not the actual activation of the Wnt

pathway.

Thanks.

Referee #3:

I have re-read the manuscript. The authors have addressed some of my comments.

The rationale behind the model is still poorly explained and would benefit from a more explicit and detailed explanation.

This should be done right when the model is first mentioned (second paragraph of section Biomechanical stress is generated inside the mandible), as I stated in my previous review. For example, the sentence in that paragraph: "Finally, the quantity of stress was determined based on the deformation and mechanical features of the mandible by establishing a "Cup" model using the finite element analysis software".

I assume that they input the observed spatial pattern of deformation and some mechanical properties to a FE model (this model uses the shape of a cup) and then the output of the model, or prediction, is the spatial pattern of stress over the cup? This needs to be spelled more explicitly and with some more detail.

2nd Revision - authors' response

18th Oct 2019

Point to point response:

Referee #2:

In this revised manuscript, the authors have addressed all my questions and most of my concerns have been answered satisfactorily. There are a couple more points that need some clarification (see below) and should be addressed. I maintain my view that the story is both interesting and important and upon the revision, the scientific rigor has been met. The manuscript enhances our understanding of how tissue mechanics contributes to tissue development and the findings also have a more general application to other epithelial organ systems. Therefore, if the couple points below can be fully addressed, I would recommend this paper to be published at EMBO.

We appreciate reviewer 2's positive feedback on our revised manuscript.

Major comments:

1. This is regarding my comment 8 in the previous review, where I asked the authors whether compressed dissociated dental follicle cells can "sustain" the expression p-ERK and RUNX2 DURING continued compression. The authors have interpreted that as how long the cell can sustain these expression AFTER compression. The answer to this is already provided in the revised manuscript as they've shown that after 4 hours of compression, p-ERK and RUNX2 staining are both starting to decrease. I thought this is

somewhat interesting, given that the expression of these proteins are maintained *in vivo*, as tissues are under constant compression. Can the authors point this out and discuss why expression cannot be maintained by continuous compression in culture? Perhaps the maintenance requires additional signals?

Thank you for the comments. Actually in this experiment, we first compressed the cells for 2 hours then removed the force (group of RFs) and cultured for another 1, 2, or 4 hours before harvesting cells. We found that both p-ERK1/2 and RUNX2 staining were increased upon loading force and were decreased at the 4th hour after removing force (Fig 4U-V). It indicates that the activation of RUNX2 and p-ERK1/2 is reversible when the force is removed on the cellular level. This finding was consistent with the result *in vivo* (Fig. 4A-L).

However, in the continued compression experiment, the expression levels of both p-ERK1/2 and RUNX2 were highest at 2nd hour, but decreased at 4th hour (Appendix Fig S8). This phenomenon may be related with the drawback of the static compression method itself. The weighted glass coverslip may prevent the infiltration of culture media into the cells after long-term pressure, which may influence the results.

2. So this could be my ignorance, but I am only aware that it is the level of the nuclear beta-catenin protein that is used as an indicator of active Wnt signaling, and not the beta-catenin RNA level in the nuclei. If there is a paper that shows differential distribution of beta-catenin RNA in nuclei vs cytoplasm in response to Wnt signaling, please provide the citation for that. Otherwise, the authors really have to show beta-catenin antibody staining, as opposed to *in situ* or RNA scope, in order to claim activated Wnt signal. Otherwise, the RNA level merely suggests that those cells are capable of Wnt signaling as beta-catenin expression is increased, and not the actual activation of the Wnt pathway.

Thank you for the comments and suggestions. In this revised manuscript, we added the additional experiments of β -catenin immunostaining. Please see the new panel in Fig 6L-N. The result showed that the nuclear β -catenin in the mesenchyme decreased, while that in the epithelium increased from E60 to E90. This finding is consistent with previous RNAscope result. The corresponding manuscript and figure legend have been revised.

Fig 6 L-N

Expression of β -catenin in the resting (E60) and initiation (E90) stages

Thanks.

Referee #3:

I have re-read the manuscript. The authors have addressed some of my comments.

The rationale behind the model is still poorly explained and would benefit from a more explicit and detailed explanation.

This should be done right when the model is first mentioned (second paragraph of section Biomechanical stress is generated inside the mandible), as I stated in my previous review. For example, the sentence in that paragraph: "Finally, the quantity of stress was determined based on the deformation and mechanical features of the mandible by establishing a "Cup" model using the finite element analysis software".

I assume that they input the observed spatial pattern of deformation and some mechanical properties to a FE model (this model uses the shape of a cup) and then the output of the model, or prediction, is the spatial pattern of stress over the cup? This needs to be spelled more explicitly and with some more detail.

We thank the reviewer for raising the comment. We have clarified the rationale of the mechanical model by stating clearly the inputs and outputs of the modeling right after the sentence the reviewer mentioned (marked in blue).

3rd Editorial Decision

11th Nov 2019

Thank you for submitting the revised manuscript and clarifying the remaining issues raised by the reviewers. I apologise for the delay in communicating the decision due to travel obligations and the

high manuscript submission rate to our office at the moment. The revised manuscript has now been seen by reviewer #2, who now recommends publication of the manuscript. However, I am afraid that there are a couple of editorial things that still have to be ironed out before I can extend a formal acceptance of the manuscript.

REFeree REPORTS:

Referee #2:

in this revised manuscript, the authors were able to satisfactorily address my remaining questions regarding 1) how cells sustain expression of key markers under continuous compression and 2) the localization of beta catenin protein in cells during resting and initiation stages. As I mentioned in my previous comments, the authors have presented a very interesting story that demonstrates how tissue compressive forces coordinate the development of epithelial structures. The evidence shown in the final manuscript is compelling. The findings are significant and of broad interest to the EMBO readership. Given that now all my questions have been addressed, I would recommend and support the publication of this paper at EMBO.

3rd Revision - authors' response

18th Nov 2019

The authors performed the requested editorial changes.

4th Editorial Decision

21st Nov 2019

Thank you for submitting the revised version of your manuscript. I am now pleased to inform you that your manuscript has been accepted for publication.

Corresponding Author Name: Songlin Wang

Journal Submitted to: The EMBO Journal

Manuscript Number: EMBOJ-2019-102374R